# Testing Most Influential Sets

**Lucas D. Konrad**[*]
Vienna University of Economics and Business
1020 Vienna, Austria
lucas.konrad@wu.ac.at

**Nikolas Kuschnig**[*]
Monash University
3145 Caulfield, Australia
nikolas.kuschnig@monash.edu

## Abstract

Small influential data subsets can dramatically impact model conclusions, with a few data points overturning key findings. While recent work identifies these *most influential sets*, there is no formal way to tell when maximum influence is excessive rather than expected under natural random sampling variation. We address this gap by developing a principled framework for most influential sets. Focusing on linear least-squares, we derive a convenient exact influence formula and identify the extreme value distributions of maximal influence — the heavy-tailed Fréchet for constant-size sets and heavy-tailed data, and the well-behaved Gumbel for growing sets or light tails. This allows us to conduct rigorous hypothesis tests for excessive influence. We demonstrate through applications across economics, biology, and machine learning benchmarks, resolving contested findings and replacing ad-hoc heuristics with rigorous inference.

## 1 Introduction

Machine learning (ML) models and statistical inferences can be highly sensitive to small subsets of data. In many applications, just a handful of samples can overturn key conclusions: two countries nullify the estimated effect of geography on development (Kuschnig et al., 2021), a single outlier flips the sign of a treatment effect (Broderick et al., 2023), or a small group of individuals drives disparate outcomes in algorithmic decision-making (Black & Fredrikson, 2021). These *most influential sets* — data subsets with the greatest influence on model predictions — are central to questions of interprtability, fairness, and robustness in modern machine learning (see, e.g., Black & Fredrikson, 2021; Chen et al., 2018; Chhabra et al., 2023; Ghorbani & Zou, 2019; Sattigeri et al., 2022).

Despite their practical importance, practitioners lack principled tools to assess whether a set's influence is genuinely problematic. Current practice relies on heuristics, ad-hoc sensitivity checks, and domain expertise, while approximate methods such as influence functions (Koh & Liang, 2017; Fisher et al., 2023; Schioppa et al., 2023) systematically underestimate the impacts of sets and extreme cases (Basu et al., 2020; Koh et al., 2019). Recent work highlights both the promises and challenges of most influential subsets — small sets can drive results even in randomized trials (Broderick et al., 2023; Kuschnig et al., 2021), heuristic algorithms can fail in simple settings (Hu et al., 2024; Huang et al., 2025), and influence bounds remain an active area of research (Moitra & Rohatgi, 2023; Freund & Hopkins, 2023; Rubinstein & Hopkins, 2025). What remains missing is a principled method to distinguish natural sampling variation from genuinely excessive influence.

We develop a statistical framework for assessing the significance of most influential sets. By focusing on linear regression — a tractable, interpretable, and widely-used setting that underlies many modern methods — we derive the exact asymptotic distributions of maximal influence. We show that two distinct regimes emerge depending on the size of the influential set: when the size is fixed, maximal influence converges to a heavy-tailed Fréchet distribution; when the size grows with the sample, maximal influence converges a well-behaved Gumbel distribution. Our results enable principled hypothesis tests for excessive influence, replacing ad-hoc diagnostics with rigorous statistical procedures. We demonstrate their practical value via applications across economics, biology, and machine learning benchmarks, resolving ambiguous cases where influential sets drive contested findings.

---

[*]Authors are listed in alphabetical order and contributed equally.

**Contributions.**   We present a comprehensive analysis of the influence of most influential sets, both theoretically and in practical applications. Our main contributions are:

1. **Theoretical foundations.** We derive distributions for the influence of most influential sets, establishing their extreme value behavior and enabling statistical testing.
2. **Efficient implementation.** We provide a closed-form formula for evaluating set influence, making our approach practical for real-world applications.
3. **Empirical validation.** We demonstrate the utility of our framework across domains, re-solving the contested 'Blessing of Bad Geography' in economics, assessing robustness in biological data of sparrow morphology, and auditing fairness in ML benchmark datasets.

To summarize — we provide the first rigorous theoretical results that allow us to *interpret influence*, and demonstrate practicality by resolving contested findings.

**Outline.**   The remainder of the paper is structured as follows. Section 2 introduces the problem of most influential sets and formalizes the setting. Section 3 presents our theoretical results on the distribution of maximal influence. Section 4 demonstrates the practical merits of our framework through simulations and empirical applications. Section 5 discusses implications, limitations, and future directions, and Section 6 concludes.

## 2   PRELIMINARIES AND BACKGROUND

Practitioners routinely encounter situations where small subsets of data points drive key conclusions. Consider the following scenarios:

- **Scientific discovery:** Rugged terrain generally hinders economic development, but not in Africa. What if this striking result is driven by just two small island nations?
- **Fairness auditing:** An algorithmic decision-making system produces different outcomes for a protected group. What if the disparity can be explained by only a handful of data points?
- **Data cleaning:** A single influential point among a thousand samples flips a strong correlation to a null result. Should we trust the original finding or the one without the outlier?
- **Data preprocessing:** A microcredit experiment shows negligible outcome variations overall, except for a few outliers. How should we prepare and analyze the sample?

At the core of these examples lie *most influential sets*, which exert disproportionate influence on an estimate or prediction. These sets are intuitive to interpret, directly tied to the quantity of interest, and provide a new dimension for assessing estimates by highlighting their support in the data.

What has been unclear, however, is *how to interpret and deal with influential sets*. Existing influence methods quantify how much changes, but offer little guidance on interpretation. Current practice relies heavily on domain expertise and ad-hoc rules, lacking a statistically rigorous framework for judging influence. Heuristics, such as sign-flips or significance thresholds, can flag effects that are stable and miss genuinely problematic influence. We address this gap by quantifying *whether observed maximum influence is statistically compatible with natural sampling variation*.

### 2.1   FORMAL PROBLEM STATEMENT

We consider a supervised learning task with input space $\mathcal{X} \subset \mathbb{R}^P$ and target space $\mathcal{Y} \subset \mathbb{R}$. The goal is to learn a function $f(\theta, \cdot) : \mathcal{X} \mapsto \mathcal{Y}$ parameterized by $\theta \in \mathbb{R}^Q$. Given training data $\{(x_n, y_n)\}_{n=1}^N$ and a loss function $\mathcal{L}(\cdot, \cdot)$, we learn parameters by solving

$$\hat{\theta} = \arg\min_{\theta \in \mathbb{R}^Q} \sum_{n=1}^N \mathcal{L}\left(f(\theta, x_n), y_n\right).$$

Let $[N] = \{1, \ldots, N\}$ and denote both an index set and its corresponding subsample as $\mathbb{S} \subset [N]$. For any subset $\mathbb{S}$, we use a subscript $\hat{\theta}_{-\mathbb{S}}$ to denote an estimate $\hat{\theta}$ without $\mathbb{S}$, i.e.

$$\hat{\theta}_{-\mathbb{S}} = \arg\min_{\theta \in \mathbb{R}^Q} \sum_{n \notin \mathbb{S}} \mathcal{L}\left(f(\theta, x_n), y_n\right).$$

**Definition** (Most Influential Set). *For a positive integer $k \ll N$, the $k$-most influential set is*

$$\mathbb{S}_k^{\max} := \underset{\mathbb{S} \subset [N], |\mathbb{S}| = k}{\arg\max} \ \Delta\left(\mathbb{S}; \phi\right),$$

*where $\Delta\left(\mathbb{S}; \phi\right) = \phi(\hat{\theta}) - \phi(\hat{\theta}_{-\mathbb{S}})$ is the* influence *of subset $\mathbb{S}$ on the scalar target function $\phi : \mathbb{R}^Q \mapsto \mathbb{R}$. We denote the maximum influence as $\Delta^{\max} = \Delta\left(\mathbb{S}_k^{\max}; \phi\right)$.*

**Research Question.** What is the probability distribution of $\Delta^{\max}$, and how can we distinguish excessive influence from natural sampling variation?

## 2.2 INFLUENCE FUNCTIONS VS. EXACT INFLUENCE

A common and related approach to study influence is via *influence functions* (Fisher et al., 2023; Hu et al., 2024; Koh & Liang, 2017). These are motivated by reweighing via the perturbation

$$\hat{\theta}(\epsilon; \mathbb{S}) := \underset{\theta \in \mathbb{R}^Q}{\arg\min} \frac{1}{N} \sum_{n=1}^{N} \mathcal{L}\left(f(\theta, x_n), y_n\right) + \epsilon \sum_{i \in \mathbb{S}} \mathcal{L}\left(f(\theta, x_i), y_i\right).$$

Setting $\epsilon = 0$ recovers $\hat{\theta}$, while $\epsilon = -N^{-1}$ yields $\hat{\theta}_{-\mathbb{S}}$. The influence function is the first-order linear approximation at $\epsilon = 0$.

While influence functions are computationally convenient, they are unreliable even for simple models (Basu et al., 2020; Hu et al., 2024; Huang et al., 2025; Koh et al., 2019). In particular, they systematically underestimate the impact of (a) *sets* of data points and (b) highly *influential* data points. This occurs because the first-order approximation cannot reflect higher-order effects from the interplay between data points or differential leverage scores.

**Exact Maximum Influence.** We therefore derive an *exact influence formula* (in Section 3) and characterize the behavior of maximum influence (Section 3.1) to enable principled testing (Section 3.3). We focus on the tractable but ubiquitous linear setting, allowing us to accurately portray most influential sets, for which extreme behavior dominates and first-order approximations fail most dramatically.

### 2.2.1 EXTREME VALUE THEORY

Our goal is to characterize the behavior of $\Delta^{\max}$, the influence of the most influential set. This quantity is defined through maximization over all possible subsets, and its distribution is governed by extreme value theory rather than classical asymptotics. See Figure 1 for an illustration. When taking maxima over random quantities, three possible limiting distributions can emerge (Fisher & Tippett, 1928; Gnedenko, 1943): the well-behaved Gumbel (Type I), the heavy-tailed Fréchet (Type II), and the bounded Weibull (Type III).

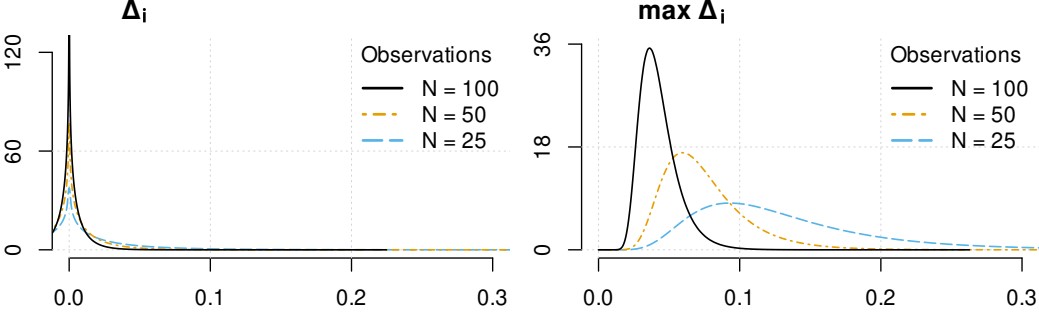

Figure 1: Illustration of the distribution of $\Delta(\{i\})$ for general observations, and the maximum influence $\Delta^{\max}$ over all possible $i$, for $N \in \{25, 50, 100\}$. One can clearly discern an upward shift and a substantial increase of the density in the tails.

We need to determine which extreme value distribution (EVD) attracts $\Delta^{\max}$. Specifically, we distinguish between the Gumbel distribution with exponential tails, and the Fréchet distribution with polynomial tails allowing for potentially arbitrarily large influence. (The Weibull distribution can be ruled out since influence is unbounded.) Once we establish the asymptotic distribution, we show how it applies to finite samples.

## 2.3 SETTING

Consider the standard linear regression model, where $f$ is a linear function relating random features $X$ to the outcome $Y$ via the parameter vector $\theta$. The ordinary least-squares (OLS) population estimator of $\theta$ is

$$\tilde{\theta} := \mathbb{E}[XX']^{-1}\mathbb{E}[XY]$$

yielding fitted values $\hat{Y} = X'\tilde{\theta}$ and the associated residuals $R = Y - X'\tilde{\theta}$. Stacking $N$ observed training samples yields the design matrix $\mathbf{X} \in \mathbb{R}^{N \times P}$ and outcome vector $\mathbf{y} \in \mathbb{R}^N$. For OLS, we assume that $\mathbf{X}'\mathbf{X}$ is invertible and remains so after removing any subset.[1] The sample OLS estimator is then

$$\hat{\theta} := \arg\min_{\theta}\|\mathbf{y} - \mathbf{X}\theta\|^2 = (\mathbf{X}'\mathbf{X})^{-1}\mathbf{X}'\mathbf{y},$$

yielding predictions $\hat{\mathbf{y}} = \mathbf{X}\hat{\theta}$ and residuals $\mathbf{r} = \mathbf{y} - \hat{\mathbf{y}}$.

For our main results, we consider least-squares estimates of the regression coefficients, with and without penalization. For illustration, however, we will consider a univariate model with one positive coefficient of interest and target function $\phi(\theta) = \theta_1$. This eases notation considerably and comes without loss of generality — theoretical results apply to more general settings.[2]

## 3 PROPOSED APPROACH

The influence of a single observation $i$ is well-known (Belsley et al., 1980; Cook, 1979; Walker & Birch, 1988) to be

$$\Delta(\{i\}) = \frac{(\mathbf{X}'\mathbf{X})^{-1}\mathbf{x}_i r_i}{1 - h_i},$$

where $h_i$ is the leverage score and $r_i$ the residual of observation $i$.

We extend this result to be (i) applicable to sets of observations, and (ii) analytically convenient.

**Proposition 1.** *The influence of some set $\mathbb{S}$ on the least-squares estimator $\hat{\theta}$ is*

$$\Delta(\mathbb{S}) = \left(\mathbf{X}'_{-\mathbb{S}}\mathbf{X}_{-\mathbb{S}} + \lambda\mathbf{I}_P\right)^{-1}\mathbf{X}'_{\mathbb{S}}\mathbf{r}_{\mathbb{S}}, \tag{1}$$

*where $\lambda \geqslant 0$ is an optional penalization parameter.*

*Proof.* Partition the full normal equations into $\mathbb{S}$ and $-\mathbb{S}$, and subtract $\mathbf{X}'_{\mathbb{S}}\mathbf{X}_{\mathbb{S}}\hat{\theta}$:

$$(\mathbf{X}^\top\mathbf{X} + \lambda\mathbf{I})\hat{\theta} = \mathbf{X}'\mathbf{y}$$

$$(\mathbf{X}'_{-\mathbb{S}}\mathbf{X}_{-\mathbb{S}} + \mathbf{X}'_{\mathbb{S}}\mathbf{X}_{\mathbb{S}} + \lambda\mathbf{I})\hat{\theta} = \mathbf{X}'_{-\mathbb{S}}\mathbf{y}_{-\mathbb{S}} + \mathbf{X}'_{\mathbb{S}}\mathbf{y}_{\mathbb{S}}$$

$$(\mathbf{X}'_{-\mathbb{S}}\mathbf{X}_{-\mathbb{S}} + \lambda\mathbf{I})\hat{\theta} = \mathbf{X}'_{-\mathbb{S}}\mathbf{y}_{-\mathbb{S}} + \mathbf{X}'_{\mathbb{S}}(\mathbf{y}_{\mathbb{S}} - \mathbf{X}_{\mathbb{S}}\hat{\theta}).$$

The key element stems from $\mathbf{r}_{\mathbb{S}} = \mathbf{y}_{\mathbb{S}} - \mathbf{X}_{\mathbb{S}}\hat{\theta}$, and the result follows from subtracting the leave-out normal equations and inverting:

$$(\mathbf{X}'_{-\mathbb{S}}\mathbf{X}_{-\mathbb{S}} + \lambda\mathbf{I})\hat{\theta} = \mathbf{X}'_{-\mathbb{S}}\mathbf{y}_{-\mathbb{S}} + \mathbf{X}'_{\mathbb{S}}\mathbf{r}_{\mathbb{S}}$$

$$(\mathbf{X}'_{-\mathbb{S}}\mathbf{X}_{-\mathbb{S}} + \lambda\mathbf{I})\left(\hat{\theta} - \hat{\theta}_{-\mathbb{S}}\right) = \mathbf{X}'_{\mathbb{S}}\mathbf{r}_{\mathbb{S}}$$

$$\Delta(\mathbb{S}) = (\mathbf{X}'_{-\mathbb{S}}\mathbf{X}_{-\mathbb{S}} + \lambda\mathbf{I})^{-1}\mathbf{X}'_{\mathbb{S}}\mathbf{r}_{\mathbb{S}}.$$

We thank an anonymous reviewer for pointing us towards this elegant proof. $\square$

---

[1]This is not necessary for our results, as they extend to ridge regression, where the penalization parameter $\lambda > 0$ in $\mathbf{X}'\mathbf{X} + \lambda\mathbf{I}$ creates a ridge that guarantees invertibility.

[2]Multiple features can be factored out via the Frisch-Waugh-Lovell theorem under mild assumptions, leaving an equivalent univariate regression. A different sign can be accommodated by simply sign-flipping the feature of interest or adjusting the target function $\phi$ accordingly.

Proposition 1 reveals the *additive structure* of individual contributions in the numerator, as well as the *multiplicative adjustment* from the denominator. It provides an exact closed-form expression that avoids re-fitting the model for each candidate subset.

## 3.1 EXTREME VALUE DISTRIBUTIONS

We now turn to finding the distribution of $\Delta(\mathbb{S})$ for the *most influential set*, $\mathbb{S}_k^{\max}$. Since this quantity is defined by an extremal operation (maximization over all possible subsets), its asymptotic behavior is governed by extreme value theory. Specifically, we seek the limiting EVD $H$ such that $\Delta^{\max} \in \text{MDA}(H)$, i.e., $\Delta^{\max}$ lies in the maximum domain of attraction of $H$.

Two canonical EVDs are of particular interest: the Fréchet (Type II) distribution $\Phi_\alpha$ for heavy-tailed variables, and the Gumbel (Type I) distribution $\Lambda$ for light-tailed variables. We distinguish two practically relevant regimes based on how the subset size $k$ scales with sample size $N$:

1. **Constant-size sets:** $k$ remains fixed as $N \to \infty$.
2. **Growing-size sets:** $k$ grows as $N \to \infty$ and $k/N \to 0$

Both regimes have been considered in practical applications (see, e.g, Broderick et al., 2023; Kuschnig et al., 2021), and — as we will show next — they yield fundamentally different asymptotic behavior with important implications for the interpretation of influence.

### 3.1.1 CONSTANT-SIZE SETS

**Theorem 1** (EVD for constant-size sets). *Suppose $\mathbb{E}\left[X^2\right] < \infty$, and that the heavier tail of $X_i, R_i$ decays at polynomial speed with coefficients $\xi_x, \xi_r$ where $\min\{\xi_x, \xi_r\} < \infty$. If $|\mathbb{S}_k^{\max}|$ remains constant as $N \to \infty$, then*

$$\lim_{N \to \infty} \Delta^{\max} \sim \text{Fréchet}(a, b, \xi),$$

*with location parameter $a$, scale parameter $b$, and shape parameter $\xi = \min\{\xi_x, \xi_r\}$.*

*Proof sketch.* Let $C := \sum_{i \in \mathbb{S}} X_i R_i$ and $D := \sum_{n=1}^N X_n^2$. Notice that $C$ and $D_{-\mathbb{S}}^{-1}$ are asymptotically independent. Since $X_i$ and $R_i$ have polynomial tails with coefficients $\xi_x, \xi_r$, their product $C \in \text{MDA}(\Phi_\xi)$, with $\xi = \min\{\xi_x, \xi_r\}$, since its upper tail behaves like the tail of $\max\{X_i R_i\}$ for $i \in \mathbb{S}_k^{\max}$. Lemma 1 shows that the inverse sum $D_{-\mathbb{S}}^{-1} \in \text{MDA}(\Lambda)$, and the product $C D_{-\mathbb{S}}^{-1}$ inherits the Fréchet behavior from $C$ by Lemma 2. □

This result shows that, for constant-size sets, $\Delta^{\max}$ inherits tail behavior from the heavier tail of $R$ and $X$. If one of them is sufficiently heavy-tailed, even small sets can exert extreme influence with non-negligible probability. Corollary 1 simplifies Theorem 1 in absence of heavy tails.

**Corollary 1.** *If the tail coefficients of both $X_i$ and $R_i$ are infinite, then*

$$\lim_{N \to \infty} \Delta^{\max} \sim \text{Gumbel}(a, b).$$

## 3.2 GROWING SETS

When the most influential set grows 'slow enough' with the sample size, the central limit theorem (CLT) dominates the asymptotic behavior:

**Theorem 2** (EVD for growing sets). *If $\{X_n R_n\}_{n=1}^N$ satisfies the conditions of a CLT and $|\mathbb{S}_k^{\max}|$ grows at $o(N)$ but faster than $O(1)$, then*

$$\lim_{N \to \infty} \Delta^{\max} \sim \text{Gumbel}(a, b).$$

*Proof sketch.* Let $m_k = |\mathbb{S}_k^{\max}|$. By assumption, $m_k \to \infty$ and $m_k = o(N)$ as $N \to \infty$. The numerator, $C$, can be written as a partial sum of $m_k$ terms and grows at rate $\mathcal{O}(m_k)$. Since $\{X_n R_n\}_{n=1}^N$ satisfies the conditions of a CLT, we obtain $(C - \mathbb{E}[C])/\sqrt{m_k} \xrightarrow{d} \mathcal{N}(\mu, \sigma^2)$ as $N \to \infty$. Hence, following Lemma 2 and Corollary 2, the product $C D_{-\mathbb{S}}^{-1}$ lies in the maximum domain of attraction of the Gumbel distribution. □

This reveals a fundamental distinction: constant-size sets are dominated by the heaviest tail, while, for growing sets, $\Delta^{\max}$ converges to a well-behaved Gumbel distribution with exponentially decaying tails. This result holds regardless of the underlying distributions of $X$ and $R$ as long as the variance of $X_i \cdot R_i$ is finite.

## 3.3 Implementation

With the theoretical results established, we turn towards practical implementation. Assuming there is a most influential set of interest,[3] our procedure follows three steps:

1. **Choose the EVD family.** Our theoretical results guide the decision between the Gumbel and Fréchet families, which based on the hypothesized set size and the tail behavior of $X$ and $R$. For the latter, we can estimate tail coefficients using maximum likelihood estimation (MLE; Smith, 1985; Bücher & Segers, 2017). If $1/\xi$ is sufficiently close to zero, we default to the Gumbel distribution (per Corollary 1 and Theorem 2). Otherwise, we use the Fréchet distribution with shape parameter $\xi$, following Theorem 1.[4]

2. **Estimate EVD parameters.** With the EVD known, we estimate its location and scale parameters $a, b$ e.g., using the block maxima method (Coles, 2002; de Haan & Ferreira, 2006). For this, we divide the sample (excluding $\mathbb{S}_k^{\max}$ for robustness) into $M$ blocks of size $N/M$, compute $\Delta^{\max}$ for each block, and use MLE based on these draws. Since selecting the maximum out of $N/M$ observations reduces the expected maximum compared to the full sample, a bias correction can be applied for the Gumbel distribution, leveraging that the densities for sizes $N$ and $N/M$ are related by

$$F^N(x) \xrightarrow{d} \text{Gumbel}(a, b) \quad \text{and} \quad [F^{N/M}(x)]^M \xrightarrow{d} \text{Gumbel}(a, b),$$

which yields the location correction $\tilde{a} = \hat{a} + b \log(M)$, where $\hat{a}$ is the MLE.

3. **Perform hypothesis test.** Finally, we test the null hypothesis $H_0$ that the observed influence reflects natural sampling variation against the alternative $H_1$ of excessive influence. Based on the estimated parameters, we can simply compute the $p$-value as $P(\Delta^{\max} \geqslant \delta_{\text{obs}})$ where $\delta_{\text{obs}}$ is the observed maximum influence.

**Computation.** Thanks to Proposition 1, our procedure is practical and computationally convenient, allowing for application to large and varied datasets. While accurately estimating Extreme Value Distributions (EVDs) is a recognized and separate challenge in its own right, the maximum likelihood steps within our framework are simple and well-behaved, optimizing over only two parameters in the Gumbel case. The primary computational constraint instead stems from finding most influential sets — we need to approximate $\Delta^{\max}$ for the $M$ block maxima estimates. For computational tractability, we use an adaptive greedy algorithm (Hu et al., 2024; Kuschnig et al., 2021) with complexity $\mathcal{O}(Mk)$ and an efficient implementation based on our closed-form influence formula for sets.

## 4 Experiments

In this section, we validate our theoretical predictions, investigate convergence in small samples, and demonstrate their practical relevance and utility through real-world applications spanning economics, biology, and machine learning.

## 4.1 Simulation Study

We begin with a controlled setting, where we (i) illustrate, (ii) investigate convergence for small samples, and (iii) evaluate empirical estimation.

---

[3]This set can be obtained with any of the methods in the literature (Broderick et al., 2023; Kuschnig et al., 2021; Freund & Hopkins, 2023), and could, e.g., be the smallest set that achieves a sign-flip — a commonly considered heuristic cutoff. If no such heuristic is used and multiple tests are considered (e.g., for different coefficients or sets sizes), a multiple testing correction should be applied. Note that EVT controls the implicit search for the most influential set over the $\binom{N}{k}$ possible subsets.

[4]Small values of $\xi$ correspond to extremely heavy-tailed distributions where the variance ($\xi \leqslant 2$) or even the mean ($\xi \leqslant 1$) become infinite. Such cases pose practical challenges for statistical inference, and imply that arbitrarily large influence is possible in our case.

**Illustration.** Figure 2 illustrates our approach on a simple linear regression with one moderately influential point due to high leverage. Panel A visualizes the data, significance thresholds (at the 10, 5, and 1% significance levels) as a function of predictor and response values. Panel B presents the underlying extreme value analysis: block maxima inform the estimated Gumbel distribution, yielding a $p$-value of 0.04 for the observation of interest.

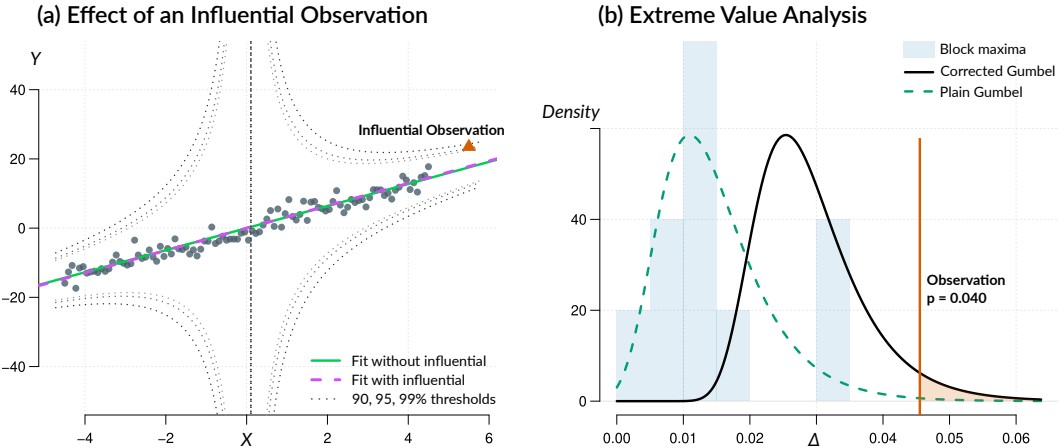

Figure 2: Illustration of our methodology on a simple linear regression with a moderately influential observation. Panel A depicts observations, estimated regression lines with and without the influential point, and conditional significance regions at the 10, 5, and 1% levels (dotted lines). Panel B illustrates the extreme value analysis: a histogram of block maxima in the background, fitted Gumbel distributions with (solid) and without (dashed) bias correction, and the resulting $p$-value for the observation of interest.

### 4.1.1 CONVERGENCE TO EXTREME VALUE DISTRIBUTIONS

Next, we verify that maximal influence converges to the predicted extreme value distributions. We consider four scenarios based on combinations of the standard Normal and $t(5)$ distributions for $X, R$. For each scenario, we simulate $1,000$ datasets of sizes between $N = 20$ and $N = 1000$, and compare the empirical parameter estimates with the theoretical prediction. Overall, we find rapid convergence, implying that our theoretical predictions are applicable with small samples.

Figure 3 shows convergence of the scenarios to the predictions, which are indicated by dashed horizontal lines. (Details are provided in Table A1 of the Appendix.) All scenarios reliably converge for moderate sample sizes. The Normal-Normal scenario is insignificantly different from Gumbel behavior ($\xi^{-1} = 0$) for $N \geqslant 50$, and the heavy-tailed cases also exhibit the predicted Fréchet behavior ($\xi^{-1} = 0.2$). Notably, the $t(5)$–Normal case converges at slower rates, likely due to the relative instability of the inverse $(\mathbf{X}'\mathbf{X})_{\mathbb{S}}^{-1}$ in small samples. Overall, the simulation results support the applicability of Theorem 1 in small samples.

**Location and Scale Estimation.** Next, we evaluate whether block maximum MLE accurately captures the location and scale parameters for empirical testing. Results are provided in Figure A1 of the Appendix. We find that bias-corrected estimation of the location parameter works well, while the scale estimate is consistent but exhibits a minor downward bias that disappears asymptotically (consistent with known limitations of the MLE; see Dombry & Ferreira, 2019). For our goal of hypothesis testing, the overall distribution and its quantiles are recovered effectively.

### 4.2 APPLICATIONS

We investigate several real-world datasets — two applications from economics and biology, and four machine learning benchmarks — and provide the first conclusive investigation of influence.

**Economic Development and Geography.** We re-examine the controversial finding that rugged terrain benefits African economies when compared to the rest of the world (Nunn & Puga, 2012).

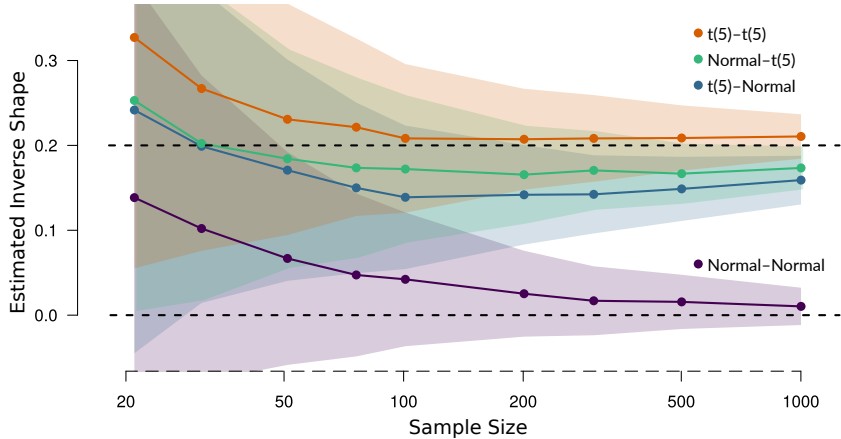

Figure 3: Convergence of empirical samples to the limiting distribution with increasing sample sizes ($N \in \{20, 30, 50, 75, 100, 150, 200, 300, 500, 1000\}$, note the non-linear scale) for four different scenarios. Dots indicate estimated means from 1000 repetitions, while the shaded area indicates $\pm 1$SD. The Normal–Normal and the $t(5)$–$t(5)$ cases quickly converge to the theoretical predictions of $\xi^{-1} = 0$ and $\xi^{-1} = 0.2$, while the mixed scenarios converge marginally slower.

Kuschnig et al. (2021) call the influence of the Seychelles, which remove significance of the estimate of interest when coupled with any of Rwanda, Lesotho, Eswatini, and the Comoros. However, lacking the statistical framework that we provide, they were not able to test whether this level of influence should be deemed excessive or not.

We can now decisively resolve this controversy. Table 1 reveals the Seychelles as excessively influential on $\hat{\theta}_{\text{rugged}}$, both individually ($p < 0.001$) and in combination with other outliers, except for Lesotho. This substantiates the suspected confounding from the size of nations (Kuschnig et al., 2021), lending statistical rigor to prior concerns and calling into question the differential relation between ruggedness and income in Africa.

Table 1: Influence of Ruggedness on log(GDP per capita in 2000)

| Influential Set | $\Delta(\mathbb{S})$ | $\hat{\hat{a}}$ | $\hat{b}$ | $p$-value |
|---|---|---|---|---|
| Seychelles | 0.077 | 0.020 | 0.004 | $< 1e^{-16}$ |
| Seychelles + Lesotho | 0.046 | 0.036 | 0.007 | 0.216 |
| Seychelles + Rwanda | 0.070 | 0.028 | 0.006 | 0.001 |
| Seychelles + Eswatini | 0.091 | 0.027 | 0.006 | $< 1e^{-16}$ |
| Seychelles + Comoros | 0.061 | 0.028 | 0.006 | 0.004 |

**Sparrow Morphology — Big Heads and Beaks.** We analyze the relation between head and tarsus length in saltmarsh sparrows, based on measurements of $N = 1,295$ sparrows with known outliers (Gjerdrum et al., 2008; Zuur et al., 2010). The baseline regression yields $\hat{\theta} = 0.011$ with a standard error of $(.030)$, implying a relation that is statistically indistinguishable from zero.

However, a curious data point moves the estimate to $0.219(.029)$, turning the estimate significantly positive. An additional data point further moves the estimate to $0.288(.032)$. These extreme impacts from a vanishing fraction of the sample are deemed excessive by our approach at any conventional significance level (both $p < 0.001$).[5]

---

[5]One possible explanation for this excessive influence are data entry errors: The first observation (an outlier in both head and tarsus size) may have the two (adjacent) features mixed up — when swapped, they would fit well into overall averages. The second observation (an outlier in one feature) stands out with both values being equal up to the one significant digit.

**Machine Learning Benchmarks.** We apply our framework to four widely-used regression benchmarks: Law School, Adult Income, Boston Housing, and Communities & Crime. For each dataset, we identify a most influential set of interest and test for excessive influence.

- **Law School** ($N = 20,800$): We examine the coefficient for the 'Other' race indicator, with 378 relevant samples. We consider two sets: 77 data points that move the estimate from $-0.0412$ (.0144) to $0.1117$ (.0159), creating a significant estimate with flipped sign, and 17 data points that reduce the estimate to $-0.0223$ (.0097). Our approach indicates that the larger set's influence falls within expected variation, while the smaller set exhibits statistically excessive influence ($p = 0.019$).

- **Adult Income** ($N = 32,561$): We investigate the top 1% most influential sets (325 points) that shift the 'Male' indicator from $\hat{\theta} = 0.062$, either raising it to $0.0992$ or decreasing it to $0.0214$. Despite these considerable shifts from a small fraction of the data, neither is deemed excessively influential by our approach.

- **Boston Housing** ($N = 506$): We focus on the effect of crime rate on house values. The baseline (highly significant) coefficient $-0.1080$ (.0329) is rendered insignificant at $-0.0352$ (.0556) after excluding just 6 observations. In this case, the underlying EVD is Fréchet with inverse shape $\xi^{-1} = 0.29$ due to the heavy tail of the crime variable. The set's influence is highly significant ($p = 0.001$), indicating excessive influence.

- **Communities & Crime** ($N = 1,994$): We investigate 2 and 2 data points with substantial influence on the relation between race and crime rates. The complete set is not extreme, as the points cancel each other out. After exclusion, the first subset of two increases the coefficient by more than 22%, which is deemed excessive $p < 0.001$. When re-estimating after their exclusion, the second set decreases the estimate by more than 10% and is deemed excessive at the 5% level ($p = 0.014$). (See Table A2 for details.)

## 5 DISCUSSION

We develop the first rigorous statistical framework for assessing when most influential sets represent genuine problems rather than natural sampling variation. By deriving the extreme value distribution of maximal influence, we allow practitioners to replace ad-hoc sensitivity checks with principled statistical decision rules.

Our key insight is how maximal influence fundamentally depends on set size and tail behavior. For constant-size sets with polynomial tails, maximal influence follows a heavy-tailed Fréchet distribution, implying that extreme influence can be arbitrarily large. For growing sets or exponential-tailed data, maximal influence converges to a well-behaved Gumbel distribution.

These results address a critical gap in interpretable machine learning. While recent work has developed methods to identify influential sets (Broderick et al., 2023; Freund & Hopkins, 2023; Hu et al., 2024), no formal theory existed to determine when their influence is excessive. Our framework provides the long-missing theoretical foundation that enables rigorous statistical inference for influential observations and sets (first discussed by Cook, 1979).

We can clarify the applicability of heuristics that are commonly used for identifying excessive influence, and provide conclusive answers when excessively influential sets are suspected (such as the Seychelles in the ruggedness example; see Kuschnig et al., 2021). The $2/\sqrt{N}$ threshold for coefficient influence (Belsley et al., 1980), e.g., is asymptotically accurate for randomly selected observations, but is too restrictive for most influential observations, where the selection procedure necessitates extreme value theory.

**Practical Recommendations.** In general, most influential sets hold valuable information for inference. Our test is deliberately *conservative*, controlling Type I errors (false claims of excessive influence) at the cost of some Type II errors (failing to detect truly excessive influence). This reflects our view that influential sets are a natural feature of data and not a problem to be eliminated. When our test identifies an *excessively influential set*, however, we recommend:

1. **Investigate mechanism.** Document the set and investigate why it differs — it may convey genuine heterogeneity, data quality issues, unobserved confounding, or important edge cases not addressed by the model.

2. **Handle appropriately.** We argue that an excessively influential set warrants separate analysis; exclusion can be considered if it reflects measurement error or outliers that are irrelevant for the pattern of interest.

3. **Report transparently.** State the set, decision, and test outcome; if conclusions hinge on the set, report both and discuss why. We advise against trimming or winsorizing to force alignment with the remaining data; these transformations create artificial data that may obscure and distort rather than illuminate underlying relations.

**Limitations and Future Work.** Our analysis focuses on linear regression — foundational for theory and modern ML methods, but limited to contexts where interpretability is valued (Rudin, 2019). Extensions to generalized linear models, tree-based methods, or non-parametric estimators require further developments. Our asymptotic arguments leverage independence between features and residuals, which may be restrictive when dependence affects influence patterns. Explicitly addressing dependence between influence components through the selected set and across sets is possible through generalizations. While simulations show that small-sample behavior quickly converges to asymptotic predictions (even at $N = 100$), further investigation of the theory-practice gap is warranted.

Practicality of our approach hinges on estimating the extreme value null, and, hence, approximating maximum influence. Finite-sample estimation of tail behavior and EVD parameters can be delicate, and improving these estimators would directly sharpen $p$-values. Two options include leveraging domain-specific information and improved bias correction methods (Dombry & Ferreira, 2019; Oorschot & Zhou, 2020). Advances in finding influential sets remain an active research area (Hu et al., 2024; Huang et al., 2025), and could substantially improve block-maxima, reduce runtime, and broaden applicability.

**Broader Implications.** Our framework enables more reliable decision-making across domains where linear models remain the method of choice. Principled tools for understanding data points that drive model behavior are crucial for building trustworthy systems. Applications span fairness assessments — where influential sets can reveal algorithmic bias — to causal inference settings, such as randomized controlled trials or quasi-experimental econometric analyses where small data subsets can fundamentally alter estimates.

Importantly, we reframe influence as a natural feature of data requiring appropriate treatment rather than a problem to be fixed. Influential sets can represent genuine heterogeneity or important edge cases that should inform model development. This perspective enables more nuanced approaches to data analysis, where information is preserved and assessed through principled statistical inference rather than discarded based on rules of thumb.

## 6 CONCLUSION

We developed a statistical framework that transforms the assessment of most influential sets from art to science. By deriving the extreme value distributions of maximal influence, we enable rigorous hypothesis testing to distinguish excessive influence from natural variation. Applications across economics, biology, and machine learning benchmarks demonstrate the practical utility of our approach.

Our method offers clear guidance to practitioners — when small sets overturn results of interest, our tests reveal whether this influence is statistically excessive. This enables more robust and transparent decision-making in settings where reliability matters, from medical trials to policy evaluation to algorithmic systems. By providing theoretical foundations for influential set analysis, this work advances both the theory and practice of interpretable machine learning.

## REPRODUCIBILITY STATEMENT

Proofs are detailed in the Appendix, datasets are from the cited sources, and code to reproduce results are available at https://github.com/konradld/testingMIS.

## STATEMENT ON LLM USE

Large language models were used to (i) aid and polish writing, (ii) discover and retrieve related work, (iii) check results for apparent mistakes.

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

## A1 LEMMA FOR THE INVERSE SUM OF SQUARES

**Lemma 1** (Asymptotic Normality of Inverse Sum of Squares). *Let $\{X_i\}_{i=1}^{\infty}$ be a sequence of independent and identically distributed (i.i.d.) random variables satisfying:*

1. *$\mathbb{E}[X_1^4] < \infty$ (finite fourth moment)*

2. *$\mathbb{E}[X_1^2] = \mu > 0$ (positive second moment)*

3. *$\mathrm{Var}(X_1^2) = \sigma^2 > 0$ (non-degenerate variance of squares)*

*Define $S_n = \sum_{i=1}^{n} X_i^2$ and $Y_n = S_n^{-1}$. Then $Y_n$ is asymptotically normal with:*

$$n^{3/2} \left( Y_n - \frac{1}{n\mu} \right) \xrightarrow{d} \mathcal{N} \left( 0, \frac{\sigma^2}{\mu^4} \right) \quad \text{as } n \to \infty.$$

*Proof.* Define the sample mean of squares $\bar{X}_n^{(2)} = n^{-1} S_n$. By the Central Limit Theorem (CLT):

$$\sqrt{n} \left( \bar{X}_n^{(2)} - \mu \right) \xrightarrow{d} \mathcal{N}(0, \sigma^2),$$

where $\mu = \mathbb{E}[X_1^2]$ and $\sigma^2 = \mathrm{Var}(X_1^2)$ (finite by $\mathbb{E}[X_1^4] < \infty$).

Consider the transformation $g(x) = x^{-1}$, which is differentiable at $x = \mu > 0$ with derivative $g'(x) = -x^{-2}$. The Delta Method gives:

$$\sqrt{n} \left( g(\bar{X}_n^{(2)}) - g(\mu) \right) \xrightarrow{d} \mathcal{N} \left( 0, \sigma^2 \cdot [g'(\mu)]^2 \right).$$

Substituting $g(\bar{X}_n^{(2)}) = (\bar{X}_n^{(2)})^{-1} = n/S_n$ and $g(\mu) = \mu^{-1}$:

$$\sqrt{n} \left( \frac{n}{S_n} - \frac{1}{\mu} \right) \xrightarrow{d} \mathcal{N} \left( 0, \sigma^2 \cdot (-\mu^{-2})^2 \right) = \mathcal{N} \left( 0, \frac{\sigma^2}{\mu^4} \right).$$

Rewriting $n/S_n = nY_n$:

$$\sqrt{n} \left( nY_n - \mu^{-1} \right) \xrightarrow{d} \mathcal{N} \left( 0, \frac{\sigma^2}{\mu^4} \right).$$

Factoring the left side:

$$\sqrt{n} \left( nY_n - \mu^{-1} \right) = n^{1/2} \cdot n \left( Y_n - \frac{1}{n\mu} \right)$$

$$= n^{3/2} \left( Y_n - \frac{1}{n\mu} \right).$$

Thus:

$$n^{3/2} \left( Y_n - \frac{1}{n\mu} \right) \xrightarrow{d} \mathcal{N} \left( 0, \frac{\sigma^2}{\mu^4} \right).$$

$\square$

## A2   LEMMATA FOR THE PRODUCT EVD

For notational simplicity let $S := \sum_{i \in \mathbb{S}} X_i \cdot R_i$ and $T := D_{-\mathbb{S}}^{-1}$, where $D = \sum_n X_n^2$. This holds for any realization $S = s \in \mathbb{R}$ and $T = t \in \mathbb{R}^+$. Further, let $\mathrm{MDA}(H)$ denote the maximum domain of attraction of an EVD $H$ where we write $Z \in \mathrm{MDA}(H)$. We specifically denote the Fréchet as $\Phi_\alpha$ and the Gumbel as $\Lambda$. We are interested in the EVD of $\Delta = S \cdot T$.

**Lemma 2.** *Let $S \in \mathrm{MDA}(\Lambda)$ and $T \in \mathrm{MDA}(\Phi_a)$ with tail-coefficient $a > 0$ and $S$ and $T$ being independent, then $\Delta(\mathbb{S}) = S \cdot T \in \mathrm{MDA}(\Phi_a)$.*

*Proof.* Recall that for Gumbel tails ($S$) the survival function decays faster than any polynomial, i.e.,

$$\mathbb{P}(S > s) \sim \exp\left(-\frac{s - \mu}{\beta}\right) \quad \text{as } s \to \infty,$$

while for the Fréchet tails ($T$) the survival function is regularly varying with index $-a$, i.e.,

$$\mathbb{P}(T > t) \sim t^{-a} L_T(t) \quad \text{as } t \to \infty,$$

where $L_T(t)$ is a slowly varying function. The density satisfies:

$$f_T(t) \sim a t^{-a-1} L_T(t) \quad \text{as } t \to \infty.$$

We are interested in the EVD of $\Delta$, i.e., $\mathbb{P}(\Delta > \delta)$. Since $\Delta = S \cdot T$ and $S, T$ are independent by assumption:

$$\mathbb{P}(\Delta > \delta) = \mathbb{P}(ST > \delta) = \int_{\mathbb{R}^+} \mathbb{P}(S > \delta/t) f_T(t) \, dt.$$

Next, we split the integral at $M > 0$:

$$\mathbb{P}(\Delta > \delta) = \underbrace{\int_0^M \mathbb{P}(S > \delta/t) f_T(t) \, dt}_{I_1} + \underbrace{\int_M^\infty \mathbb{P}(S > \delta/t) f_T(t) \, dt}_{I_2}.$$

For fixed $M$, we have $I_1 \to 0$, as $\delta \to \infty$ since $\delta/t \to \infty$ and Gumbel tails decay faster than any polynomial, and the dominant term is $I_2$. Substitute $u = \delta/t$ ($t = \delta/u$, $dt = -(\delta/u^2) \, du$), and we have

$$I_2 = \int_M^\infty \mathbb{P}(S > \delta/t) f_T(t) \, dt = \int_0^{\delta/M} \mathbb{P}(S > u) f_T(\delta/u) \frac{\delta}{u^2} \, du.$$

Using the asymptotic form of $f_T$:

$$f_T(\delta/u) \sim a(\delta/u)^{-a-1} L_T(\delta/u),$$

we obtain

$$I_2 \sim \int_0^{\delta/M} \mathbb{P}(S > u) \left[ a\left(\frac{\delta}{u}\right)^{-a-1} L_T\left(\frac{\delta}{u}\right) \right] \frac{\delta}{u^2} \, du$$

$$= a\delta^{-a} \int_0^{\delta/M} \mathbb{P}(S > u) u^{a-1} L_T\left(\frac{\delta}{u}\right) \, du.$$

As $\delta \to \infty$, by Lemma 4 in Appendix A3, we obtain

$$\int_0^{\delta/M} \mathbb{P}(S > u) u^{a-1} L_T\left(\frac{\delta}{u}\right) \, du \sim L_T(\delta) \int_0^\infty \mathbb{P}(S > u) u^{a-1} \, du. \tag{A2.1}$$

The integral converges because:

1. near $u = 0$ we have $\mathbb{P}(S > u) \approx 1$ and $u^{a-1}$ is integrable for $a > 0$, and

2. as $u \to \infty$ the Gumbel decay dominates $u^{a-1}$.

Denote the constant

$$C(a, S) = \int_0^\infty \mathbb{P}(S > u) u^{a-1} \, \mathrm{d}u \in (0, \infty),$$

then

$$\mathbb{P}(\Delta > \delta) \sim a\delta^{-a} L_T(\delta) C(a, S) = \delta^{-a} \left( aC(a, S) L_T(\delta) \right).$$

The term in parentheses is slowly varying in $\delta$ since $L_T(\delta)$ is slowly varying.

Thus, the survival function $\mathbb{P}(\Delta > \delta)$ is regularly varying with index $-a$, and therefore, $\Delta$ has Fréchet tails with tail-coefficient $a$, which concludes the proof. □

**Corollary 2.** *Following Lemma 2 and assuming a tail coefficient $a = \infty$ it follows that $S \sim$ Gumbel and thus $\Delta(\mathbb{S}) = S \cdot T \in \mathrm{MDA}(\Lambda)$.*

*Proof.* The result follows directly from properties of the Fréchet distribution. □

**Lemma 3.** *If $S \in \mathrm{MDA}(\Phi_a)$ and $T \in \mathrm{MDA}(\Phi_b)$ then $\Delta(\mathbb{S}) \in \mathrm{MDA}(\Phi_{\min\{a,b\}})$.*

*Proof.* The proof of this follows directly from Lemma 1.3.1 on the convolution closure of distribution functions with regularly varying tails in Embrechts et al. (1997). □

**Corollary 3** (Conditional EVD)**.** *Further, if $S \in \mathrm{MDA}(E)$ for some EVD $E$, it holds that*

$$\Delta(\mathbb{S}) \mid X_{-\mathbb{S}} \in \mathrm{MDA}(E).$$

## A3 LEMMA FOR ASYMPTOTIC EQUIVALENCE

**Lemma 4** (Asymptotic Equivalence Statement)**.**

$$\int_0^{\delta/M} \mathbb{P}(S > u)u^{a-1}L_T\left(\frac{\delta}{u}\right) \mathrm{d}u \sim L_T(\delta)\int_0^\infty \mathbb{P}(S > u)u^{a-1}\,\mathrm{d}u,$$

*where $S$ has Gumbel tails, $L_T$ is slowly varying, $a > 0$ is the tail coefficient, $M > 0$ is a fixed constant.*

*Proof.* For clarity, we prove this result in five steps.

STEP 1: INTEGRAL SPLITTING

Define

$$I(\delta) = \int_0^{\delta/M} \mathbb{P}(S > u)u^{a-1}L_T\left(\frac{\delta}{u}\right) \mathrm{d}u = I_1(\delta) + I_2(\delta),$$

where

$$I_1(\delta) = \int_0^1 \mathbb{P}(S > u)u^{a-1}L_T\left(\frac{\delta}{u}\right) \mathrm{d}u,$$

$$I_2(\delta) = \int_1^{\delta/M} \mathbb{P}(S > u)u^{a-1}L_T\left(\frac{\delta}{u}\right) \mathrm{d}u.$$

STEP 2: ANALYSIS OF $I_1(\delta)$ (BOUNDED DOMAIN)

For $u \in (0, 1]$, we have:

$$\lim_{\delta\to\infty} \frac{I_1(\delta)}{L_T(\delta)} = \lim_{\delta\to\infty} \int_0^1 \mathbb{P}(S > u)u^{a-1}\frac{L_T(\delta/u)}{L_T(\delta)} \mathrm{d}u$$

$$= \int_0^1 \mathbb{P}(S > u)u^{a-1}\,\mathrm{d}u,$$

by the Dominated Convergence Theorem (DCT):

- *Pointwise convergence:* for fixed $u > 0$, $\lim_{\delta\to\infty} \frac{L_T(\delta/u)}{L_T(\delta)} = 1$.

- *Dominating function:* by Potter's theorem, for any $\delta > 0$, there exists $C_\delta > 0$ such that

$$\left|\frac{L_T(\delta/u)}{L_T(\delta)}\right| \leqslant C_\delta u^{-\delta} \quad \text{for all large } \delta.$$

Choose $\delta < a$ such that $u^{a-1-\delta}$ is integrable on $(0, 1]$, then

$$\left|\mathbb{P}(S > u)u^{a-1}\frac{L_T(\delta/u)}{L_T(\delta)}\right| \leqslant C_\delta u^{a-1-\delta} \quad (\text{since } \mathbb{P} \leqslant 1),$$

and the dominating function $C_\delta u^{a-1-\delta}$ is integrable over $(0, 1]$ for $a > \delta > 0$.

STEP 3: ANALYSIS OF $I_2(\delta)$ (GROWING DOMAIN)

For $u \in [1, \delta/M]$, we have

$$\lim_{\delta\to\infty} \frac{I_2(\delta)}{L_T(\delta)} = \lim_{\delta\to\infty} \int_1^{\delta/M} \mathbb{P}(S > u)u^{a-1}\frac{L_T(\delta/u)}{L_T(\delta)} \mathrm{d}u$$

$$= \int_1^\infty \mathbb{P}(S > u)u^{a-1}\,\mathrm{d}u \quad \text{by the DCT.}$$

- *Pointwise convergence:* for fixed $u \geqslant 1$, $\lim_{\delta\to\infty} \frac{L_T(\delta/u)}{L_T(\delta)} = 1$

- *Dominating function:* by Potter's theorem, for $\delta > 0$:

$$\left| \frac{L_T(\delta/u)}{L_T(\delta)} \right| \leqslant C_\delta u^\delta \quad \text{for all large } \delta, u \geqslant 1.$$

Choose $\delta$ such that $k = a - 1 + \delta > 0$, and

$$\int_1^\infty \mathbb{P}(S > u) u^k \, \mathrm{d}u < \infty,$$

since the Gumbel decay dominates. Then

$$\left| \mathbb{P}(S > u) u^{a-1} \frac{L_T(\delta/u)}{L_T(\delta)} \right| \leqslant C_\delta \mathbb{P}(S > u) u^k,$$

and the dominating function $C_\delta \mathbb{P}(S > u) u^k$ is integrable over $[1, \infty)$.

- *Tail control:* as $\delta \to \infty$, the upper limit $\delta/M \to \infty$ and

$$\int_{\delta/M}^\infty C_\delta \mathbb{P}(S > u) u^k \, \mathrm{d}u \to 0.$$

STEP 4: NEGLIGIBILITY OF OMITTED TAIL

The tail beyond $\delta/M$ is negligible:

$$R(\delta) = \int_{\delta/M}^\infty \mathbb{P}(S > u) u^{a-1} L_T\left(\frac{\delta}{u}\right) \, \mathrm{d}u.$$

- For $u \geqslant \delta/M$, we have $\delta/u \leqslant M$, which is bounded on compact sets: $L_T(\delta/u) \leqslant C_M$.

- By the Gumbel tail properties, there exist a $\theta > 0$ s.t. $\mathbb{P}(S > u) \leqslant e^{-u^\theta}$ for large $u$. Thus

$$|R(\delta)| \leqslant C_M \int_{\delta/M}^\infty e^{-u^\theta} u^{a-1} \, \mathrm{d}u = o(1) \quad \text{as } \delta \to \infty.$$

- Since $L_T(\delta) \to \infty$ or is slowly varying, $R(\delta) = o(L_T(\delta))$

STEP 5: FINAL COMBINATION

Combining all results, we have

$$\begin{aligned}
\frac{I(\delta)}{L_T(\delta)} &= \frac{I_1(\delta) + I_2(\delta) + R(\delta)}{L_T(\delta)} \\
&= \frac{I_1(\delta)}{L_T(\delta)} + \frac{I_2(\delta)}{L_T(\delta)} + o(1) \\
&\implies \int_0^1 \mathbb{P}(S > u) u^{a-1} \, \mathrm{d}u + \int_1^\infty \mathbb{P}(S > u) u^{a-1} \, \mathrm{d}u \\
&= \int_0^\infty \mathbb{P}(S > u) u^{a-1} \, \mathrm{d}u.
\end{aligned}$$

$\square$

## A4    AUXILIARY MATERIAL

| $N$ | Distribution | Mean | Std.Dev. | Q25 | Median | Q75 |
|---|---|---|---|---|---|---|
| 20 | Normal–Normal | 0.1385 | 0.2541 | −0.0206 | 0.1293 | 0.2902 |
| | $t(5)$–Normal | 0.2417 | 0.2865 | 0.0863 | 0.2329 | 0.3994 |
| | Normal–$t(5)$ | 0.2529 | 0.2479 | 0.0910 | 0.2505 | 0.4089 |
| | $t(5)$–$t(5)$ | 0.3271 | 0.2719 | 0.1601 | 0.3144 | 0.4922 |
| 30 | Normal–Normal | 0.1021 | 0.1803 | −0.0079 | 0.1101 | 0.2291 |
| | $t(5)$–Normal | 0.1989 | 0.1845 | 0.0917 | 0.1994 | 0.3067 |
| | Normal–$t(5)$ | 0.2023 | 0.1846 | 0.0817 | 0.2101 | 0.3225 |
| | $t(5)$–$t(5)$ | 0.2670 | 0.1912 | 0.1453 | 0.2668 | 0.3961 |
| 50 | Normal–Normal | 0.0668 | 0.1255 | −0.0153 | 0.0631 | 0.1466 |
| | $t(5)$–Normal | 0.1708 | 0.1304 | 0.0890 | 0.1701 | 0.2532 |
| | Normal–$t(5)$ | 0.1843 | 0.1286 | 0.1015 | 0.1797 | 0.2704 |
| | $t(5)$–$t(5)$ | 0.2307 | 0.1363 | 0.1420 | 0.2305 | 0.3203 |
| 75 | Normal–Normal | 0.0474 | 0.0960 | −0.0163 | 0.0485 | 0.1176 |
| | $t(5)$–Normal | 0.1500 | 0.1006 | 0.0821 | 0.1469 | 0.2173 |
| | Normal–$t(5)$ | 0.1735 | 0.1057 | 0.1037 | 0.1749 | 0.2434 |
| | $t(5)$–$t(5)$ | 0.2214 | 0.1046 | 0.1561 | 0.2213 | 0.2885 |
| 100 | Normal–Normal | 0.0422 | 0.0788 | −0.0115 | 0.0429 | 0.0959 |
| | $t(5)$–Normal | 0.1389 | 0.0846 | 0.0790 | 0.1369 | 0.1948 |
| | Normal–$t(5)$ | 0.1721 | 0.0865 | 0.1183 | 0.1731 | 0.2291 |
| | $t(5)$–$t(5)$ | 0.2083 | 0.0876 | 0.1492 | 0.2056 | 0.2639 |
| 200 | Normal–Normal | 0.0253 | 0.0506 | −0.0075 | 0.0253 | 0.0582 |
| | $t(5)$–Normal | 0.1418 | 0.0587 | 0.1020 | 0.1403 | 0.1833 |
| | Normal–$t(5)$ | 0.1655 | 0.0576 | 0.1272 | 0.1632 | 0.2013 |
| | $t(5)$–$t(5)$ | 0.2072 | 0.0595 | 0.1691 | 0.2054 | 0.2450 |
| 300 | Normal–Normal | 0.0169 | 0.0406 | −0.0096 | 0.0180 | 0.0455 |
| | $t(5)$–Normal | 0.1423 | 0.0461 | 0.1122 | 0.1446 | 0.1731 |
| | Normal–$t(5)$ | 0.1705 | 0.0463 | 0.1377 | 0.1695 | 0.2006 |
| | $t(5)$–$t(5)$ | 0.2082 | 0.0511 | 0.1757 | 0.2082 | 0.2406 |
| 500 | Normal–Normal | 0.0157 | 0.0319 | −0.0049 | 0.0154 | 0.0381 |
| | $t(5)$–Normal | 0.1488 | 0.0378 | 0.1245 | 0.1499 | 0.1739 |
| | Normal–$t(5)$ | 0.1667 | 0.0354 | 0.1419 | 0.1662 | 0.1903 |
| | $t(5)$–$t(5)$ | 0.2087 | 0.0385 | 0.1833 | 0.2085 | 0.2323 |
| 1000 | Normal–Normal | 0.0103 | 0.0220 | −0.0042 | 0.0109 | 0.0243 |
| | $t(5)$–Normal | 0.1591 | 0.0287 | 0.1408 | 0.1589 | 0.1771 |
| | Normal–$t(5)$ | 0.1734 | 0.0252 | 0.1565 | 0.1743 | 0.1905 |
| | $t(5)$–$t(5)$ | 0.2105 | 0.0262 | 0.1938 | 0.2101 | 0.2273 |

Table A1:  Inverse shape estimates for different distributions and sample sizes. They indicate fast convergence to asymptotic predictions (0.0 for the Normal-Normal, and 0.2 for other cases) — for all settings, the predicted value is contained between the 25[th] and 75[th] quantile. Estimates are based on 1,000 repetitions.

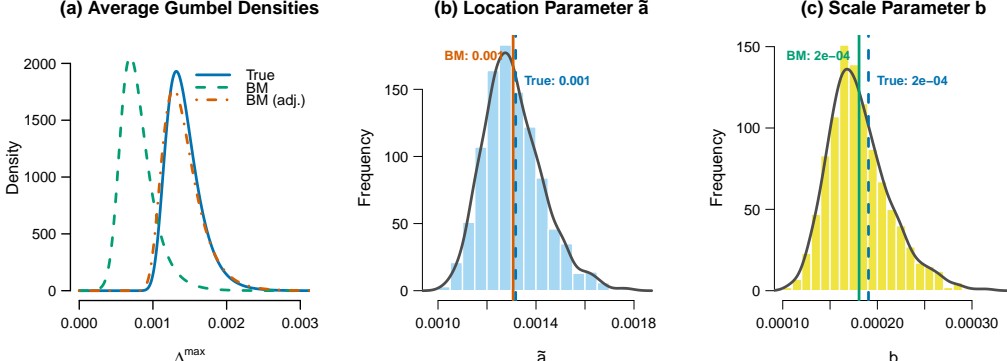

Figure A1: Simulation exercise for the performance of the simple MLE based on block maxima, correcting for block size. While the corrected location parameter $\hat{\tilde{a}}$ is close to unbiased, the scale parameter $\hat{b}$ suffers some downward bias using simple block maxima, which is in line with Dombry & Ferreira (2019). However, for practical purposes the block maxima are expected to be fitting reasonably well, as visible in panel (a).

| Set Composition | Set Size | $\Delta(\mathbb{S})$ | $\hat{\tilde{a}}$ | $\hat{b}$ | $p$-value |
|---|---|---|---|---|---|
| Full set | 4 | 0.0214 | 0.0076 | 0.0029 | 0.4914 |
| 1st partial | 2 | 0.0456 | 0.0050 | 0.0021 | $7.62e^{-7}$ |
| 2nd after excl. 1st | 2 | $-0.0241$ | 0.0051 | 0.0022 | 0.0141 |

Table A2: Influence of % Black Population on Violent Crimes. The table summarizes the results for testing the preselected set and its subsets for significant influence of the percent of black population on the violent crimes committed per population.

