# OpenReview forum: "Testing Most Influential Sets"
_ICLR.cc/2026/Conference — ICLR 2026 Poster_

### Official Review · Reviewer_PJQ8 · 2025-10-14

**Soundness:** 3
**Presentation:** 2
**Contribution:** 2
**Rating:** 2
**Confidence:** 5

**Summary:**

This paper develops a statistical framework to distinguish natural sampling variation from excessive influence in most influential subsets. The authors derive exact expressions for subset influence in univariate linear regression and establish that the maximal influence follows asymptotic extreme-value distributions (Fréchet or Gumbel). Building on this, they propose a hypothesis testing procedure based on MLE (to estimate the EVD parameters) and an adaptive greedy algorithm from prior work (to estimate maximum influence). Empirical studies on economics, biology, and classic ML datasets illustrate the method’s interpretability and robustness benefits.

**Strengths:**

- Neat theoretical framework, which elegantly links subset influence to extreme-value asymptotics

- Comprehensive empirical evaluations across diverse domains

**Weaknesses:**

- The theory focuses exclusively on 1D OLS, which is too restrictive. Based on my understanding, it is very difficult (if not impossible) to generalize Proposition 1 to high-dimensional OLS. All empirical studies also concentrate on tabular data and linear models, further limiting the paper’s applicability to modern ML settings.

- ML practitioners typically treat the training dataset as fixed rather than as a random sample from an underlying distribution. The authors should better motivate why testing statistical significance rather than simply detecting influence is meaningful in this context.

- The paper would benefit from clearer exposition of extreme-value theory and stronger intuition for the Gumbel vs. Fréchet distinction, which are currently introduced with limited background or explanation.

In summary, while the work’s statistical rigor is admirable, its theoretical contribution does not quite meet the bar for a top-tier ML conference, and its relevance to representation learning is limited. I encourage the authors to further develop and generalize the theory and consider submitting to a statistics or econometrics venue, where inferential rigor is the primary focus.

**Questions:**

I don't have further questions at this point, but would be willing to reconsider my rating if the authors can provide a clear and technically plausible pathway for extending their theoretical framework beyond the 1D OLS setting.

---

> ### Author Response · Authors · 2025-11-19
> **Summary Response**
>
> ## General Response
> We thank the reviewers for their constructive engagement with our work. We are encouraged by the consensus on theoretical rigor, novelty, and importance of addressing the long-standing challenge of assessing influential sets.
>
> **Core contribution.** Several reviewers asked us to clarify what our framework adds beyond existing influence quantification methods. The answer is fundamental: existing methods can identify influential sets but *cannot determine if their influence is excessive*.
> Prior work relies on ad-hoc heuristics (sign-flips, arbitrary thresholds like $2\sqrt{N}$, percentage rules) without theoretical foundation. Our contribution derives the distribution of maximum influence, enabling the first principled hypothesis tests to distinguish excessive influence from natural sampling variation. This transforms influential set analysis from heuristic sensitivity checking into rigorous statistical inference.
>
> Our revision addresses three main themes: (1) **scope and generalizability**, (2) **practical guidance and workflow**, and (3) **clarity of exposition**. We detail related revisions next, and follow with individual point-by-point responses to concerns raised by reviewers in the reply to this summary.
> ### Generalization beyond univariate OLS.
> We substantially expanded our theoretical framework.
> Proposition 1 now applies to *multivariate* least-squares estimators with arbitrary $P$-dimensional parameter vectors, encompassing both OLS ($\lambda = 0$) and ridge regression ($\lambda > 0$). Ridge regression ensures invertibility in high-dimensional settings where $P \gg N,$ covering practical ML applications.
> All *empirical applications* use the multivariate framework; the univariate case serves purely as *illustration* to build intuition. Full technical details appear in Appendix A1.
> ### Practical guidance and workflow.
> We enhanced practical guidance to address concerns about applicability:
>
> - **Convergence:** New simulations show reliable convergence at $N=100$  (and $N=20$ for light-tailed cases).
> - **Workflow:** We now cover the workflow before and after our test. *Before testing,* we clarify set selection and when multiple testing corrections apply. *After detection* of excessive influence, we provide guidance on investigation and responses.
> - **Why testing matters:** We clarified that our contribution provides theoretically-founded decision rules to replace ad-hoc heuristics. - Existing methods quantify influence; we enable principled decisions on whether that influence is excessive.
> ### Clarity and presentation.
> We improved exposition by adding background information on extreme value theory (including intuition and an illustrative figure), streamlining the influence function discussion and moved up key citations, fixed notational issues, and restructured for logical flow.

---

> > ### Author Response · Authors · 2025-11-19
> > **Individual Response to Reviewer PJQ8**
> >
> > ## Response to Reviewer PJQ8
> > We thank the reviewer for recognizing our "neat theoretical framework" and "comprehensive empirical evaluations." We truly appreciate the explicit offer to reconsider if we demonstrate generalization beyond 1D OLS. Our substantial revisions directly address this and all other concerns, and we respectfully request reconsideration.
> >
> > ### W1: Generalization
> > We appreciate the concern that generalizing Proposition 1 is "very difficult (if not impossible)" – we agree with the difficulty, but are happy to report we have achieved the generalization.
> >
> > The revised paper includes a new Proposition 1 (with full proof in Appendix A1):
> > $$
> >     \Delta(\mathcal{S}) = \left( \mathbf{X}_{-\mathcal{S}}' \mathbf{X}_{-\mathcal{S}} + \lambda\mathbf{I}_P \right)^{-1} \mathbf{X}_{\mathcal{S}}' \mathbf{r}_{\mathcal{S}},
> > $$
> >
> > where $\lambda \geq 0$ enables ridge regression for high-dimensional settings and multivariate settings are accommodated.
> >
> > The proof relies on the inverse matrix lemma, updating rules, and matrix calculus – arguably making it straightforward but tedious. We provide a sketch with the one-dimensional case for intuition and full details in the Appendix.
> >
> > All applications demonstrate the multivariate framework. They are limited to tabular data and linear models, which are ubiquitous in modern ML and important in their own right. We want to emphasize that analyses such as ours must start somewhere. The paper mentions extensions to other settings as important future work – we believe our work can be foundational for them.
> >
> > ### W2: Fixed datasets vs. random sampling
> > This is a thoughtful conceptual question about the frequentist vs. finite-population perspective.
> >
> > In standard regression, data is considered given and testing occurs via the random part (residuals). In our case, the data – especially the tested set – is part of the problem and cannot be taken as given.
> >
> > The frequentist perspective (our framework) is standard in ML for generalization (training data is one sample; we care about population performance) and underlies virtually all statistical inference in applied work (clinical trials, A/B tests, surveys). This perspective enables principled decision-making: "Would this influence level occur naturally if we resampled?" is actionable, while "This specific dataset has this specific property" cannot inform decisions out-of-sample. Without a distribution, we cannot distinguish genuine problems from sampling artifacts.
> >
> > **Alternative perspectives** are possible within our framework:
> >
> > - **Bayesian:** One could compute posterior probabilities of excessive influence. Our results would inform this approach, but developing it as well is beyond the scope of the paper.
> > - **Finite population:** If the dataset truly exhausts the population of interest, descriptive influence measures suffice. However, this assumption is extremely rare in statistical theory, and rarely (if ever) holds in ML applications: we typically care about future data, and not just the training set.
> >
> > ### W3: Clearer exposition of extreme-value theory
> > We fully agree and have added substantial EVT background. Before technical results, we now provide intuition and illustration for why maxima behave differently from averages. We clarify why our problem requires EVT and introduce the three possible EVD types.
> >
> > ### Venue appropriateness
> > We respectfully disagree. While our theoretical tools draw from statistics and applications span domains, the problem is fundamentally about interpretable and robust machine learning. Influential sets are central to fairness, interpretability, and robustness, and minuscule sets are known to impact even large language models (Souly et al., 2025).
> > Finally, we hope that our proof of a crucial proposition that the reviewer deemed "very difficult (if not impossible)" highlights our theoretical contribution.
> >
> > ### Summary
> > The reviewer offered to reconsider given "a clear and technically plausible pathway for extending their theoretical framework beyond the 1D OLS setting." We have provided exactly this and directly address all three weaknesses raised.
> >
> > Based on these substantial improvements, combined with the reviewer's recognition of our theoretical elegance and comprehensive empirics, we respectfully request reconsideration. Our work addresses a fundamental ML problem – distinguishing excessive maximum influence from natural influence using rigorous theory with broad practical applicability.## Response to Reviewer PJQ8
> > We thank the reviewer for recognizing our "neat theoretical framework" and "comprehensive empirical evaluations." We truly appreciate the explicit offer to reconsider if we demonstrate generalization beyond 1D OLS. Our substantial revisions directly address this and all other concerns, and we respectfully request reconsideration.

---

> ### Comment · Reviewer_PJQ8 · 2025-11-19
>
> Thanks for the response.
>
> I apologize that I didn't make it sufficiently clear about "generalization of prop 1". Deriving such a formula is clearly possible. In fact, here's a simple derivation that proves prop 1 in 4 lines without relying on induction or complex calculations or matrix inversion formula:
>
> $$(X_{-S}^\top X_{-S} + \lambda I + X_S^\top X_S) \hat\theta = X_{-S}^\top y_{-S} + X_{S}^\top y_S$$
> $$(X_{-S}^\top X_{-S} + \lambda I) \hat\theta = X_{-S}^\top y_{-S} + X_S^\top (y_S - X_S\hat\theta) = X_{-S}^\top y_{-S} + X_S^\top r_S$$
> $$(X_{-S}^\top X_{-S} + \lambda I) \hat\theta = (X_{-S}^\top X_{-S} + \lambda I) \hat\theta_{-S} + X_S^\top r_S$$
> $$\Delta_S = (X_{-S}^\top X_{-S} + \lambda I)^{-1} X_S^\top r_S$$
>
> What I really meant in my original review is: does there exist an efficient algorithm of computing subset influence in high dimension OLS. Using the above formula still requires inverting a matrix, so it does not bring benefits compared to the simple baseline: directly computing $\theta_{-S}$, then taking the difference. I think the claim *"This representation enables efficient computation without explicitly recalculating leverage scores for each subset, making our approach computationally tractable for large datasets"* in the updated draft is quite misleading.
>
> Therefore, while I appreciate the authors' efforts in working out the calculations, I still feel this is not a strong technical contribution.

---

> > ### Author Response · Authors · 2025-11-19
> >
> > We appreciate the elegant alternative derivation of our Proposition 1, and would be happy to incorporate it.
> >
> > Before we answer your *question on computational complexity*, we want to *note two important points*:
> >
> > 1. The **focus on Proposition 1 misses our main contribution** – the *derivation of the distribution of maximal influence*, providing the first principled solution to a fundamental problem. Proposition 1 enables the stochastic analysis that goes into our **Theorems 1 and 2**, and none of our analysis relies on a computationally efficient algorithm.
> > 2. We note your **concern has shifted from generalizability to efficiency**. The original review stated: *"The theory focuses exclusively on 1D OLS, which is too restrictive. Based on my understanding, it is very difficult (if not impossible) to generalize Proposition 1 to high-dimensional OLS."* We *provided exactly this generalization*, accommodating high-dimensional cases and penalized LS. Now the concern is computational efficiency.
> >
> > To address your question *"does there exist an efficient algorithm of computing subset influence in high dimension OLS"?* **Yes.** Proposition 1 shows we can leverage the known inverse $(X'X + \lambda I)^{-1},$ and update by removing the set $S.$ Via the inverse matrix lemma, we avoid the costly $\mathcal{O}(N^3)$ inversion, simply computing matrix products at complexity $\mathcal{O}(N^2 \lvert S \rvert),$ where $\lvert S \rvert \ll N,$ bringing considerable gains over directly recomputing $\hat\theta.$ Our claim of "efficient computation [...] for large datasets," is correct, and we will clarify this further in the revision.
> >
> > **To summarize:** We have conclusively addressed both your original concern (generalization) and this follow-up (efficient algorithms exist). We hope you will reconsider your evaluation in light of these substantive responses to both rounds of concerns.

---

> > > ### Comment · Reviewer_PJQ8 · 2025-11-19
> > >
> > > Sorry, I didn't follow your argument. Prop 1 shows that $\Delta(S) = (X_{-S}^\top X_{-S} + \lambda I)^{-1} X_S^\top r_S$, which involves exactly the same matrix inversion $(X_{-S}^\top X_{-S} + \lambda I)^{-1}$ if we compute $\hat \theta_{-S}$ directly. The inverse matrix formula can be applied equally to these two approaches. I don't see how Prop 1 brings computational gains and how it can leverage the known quantity $(X^\top X + \lambda I)^{-1}$.

---

> > > > ### Author Response · Authors · 2025-11-20
> > > >
> > > > Thanks for the quick response – we see the confusion now and want to clarify three points:
> > > >
> > > > **1. Our main contribution is analytical**. Proposition 1 enables the stochastic analysis underlying Theorems 1 and 2 – our core contribution of deriving the EVD of maximum influence. Our applications handle high-dimensional settings ($N > 10,000$ and $P > 100) with negligible computation time. (We can provide timings, but this has never constrained our framework.)
> > > >
> > > > **2. We have addressed your original concern**. Our revision directly provides the requested *"clear and technically plausible pathway for extending [...] beyond the 1D OLS setting."* We understand the discussion has now shifted to
> > > > computational efficiency, which we address next.
> > > >
> > > > **3. On computational efficiency.**
> > > > Perhaps your concern relates to *most influential sets in general*. Finding the most influential set among the combinatorial $N \choose \lvert S \rvert$ options is generally *computationally infeasible*. We **bypass this constraint** by focusing on the *maximum influence* itself, which we can characterize via extreme value theory.
> > > >
> > > > Regarding empirical implementation and the *re-use of the original inverse*: You are correct that Proposition 1 itself brings no computational gains – **implementation via the inverse matrix lemma** (i.e., the Sherman-Morrison-Woodbury formula) can leverage the known original quantity to bring complexity of removing set $S$ from $\mathcal{O}(N^3)$ to $\mathcal{O}(N^2 \lvert S \rvert)$ where $\lvert S \rvert \ll N.$
> > > > Let $A = (X'X + \lambda I)^{-1}$ be the known quantity and $X_S$ be the set to remove; then
> > > > $$
> > > > A_{-S}^{-1} = A^{-1} + A^{-1} X_S' (I - X_S A^{-1} X_S')^{-1} X_S A^{-1},
> > > > $$
> > > > and we merely need to perform matrix multiplications with $X_S.$ This is an arguably important implementation detail, but not our theoretical contribution.
> > > >
> > > > **To summarize:** We've directly addressed your original technical concern. Our core contribution – the first principled hypothesis test for excessive influence – stands as rigorous theory with demonstrated practical applicability. We respectfully request reconsideration based on addressing both the original concern and this follow-up question.

---

> ### Comment · Reviewer_PJQ8 · 2025-11-20
>
> My understanding is that prop 1 is not merely a theoretical results; it is a key component of the hypothesis testing procedure. The draft explicitly states that “Thanks to Proposition 1, our procedure is computationally convenient”, and the emphasis on computational efficiency appears repeatedly throughout both the paper and the rebuttal. However, after further clarification, this claim does not seem to hold.
>
> I am willing to raise my score to a 4, but this is the highest I can reasonably justify. In my view, the scope of the paper is limited (this is of course a subjective judgment), and the work requires substantial revision to accurately contextualize and frame its contributions (including but not limited to computational efficiency). In its current form, I cannot recommend acceptance.

---

> > ### Author Response · Authors · 2025-11-20
> >
> > Thank you for considering a score increase to 4.
> >
> > **On computational efficiency:** Proposition 1 does have practical implications, and as we've clarified – **our procedure is computationally efficient**. We mention this in the paper because it matters for implementation. However, we want to be clear: computational efficiency has never been framed as our primary theoretical contribution – it benefits implementation.
> >
> > Regarding the claim that computational efficiency *"appears repeatedly throughout the rebuttal"*: a search for "computation" shows this topic **appears *exclusively* in our responses to your questions** about it. In the paper itself, we mention it briefly as a practical consideration, not as a core contribution.
> >
> > **On the scope and contributions:** We respectfully maintain that we have directly and conclusively addressed your stated concerns:
> >
> > 1. **Generalization beyond 1D OLS** (your stated reason for potential reconsideration) is achieved via the multivariate framework covering arbitrary $P$-dimensional parameters and ridge regression
> > 2. **Computational efficiency** (your follow-up concern) is clarified both theoretically and practically.
> >
> > Our core contribution remains **the first principled hypothesis test for excessive influence,** which transforms influential set analysis from ad-hoc heuristics into rigorous statistical inference. This addresses a fundamental 45-year-old problem with broad applicability across ML interpretability, fairness, and robustness.
> >
> > We understand scope assessments involve subjective judgment. However, we would welcome any specific remaining concerns, as we are committed to making any revisions that would support acceptance.

---

> > ### Author Response · Authors · 2025-11-26
> > **Addressing scope concerns**
> >
> > ## Addressing Scope Concerns
> > Multiple reviewers raised concerns about the scope of our framework, so we sketch the **extension of our results to M-estimators**, including logistic regression and GLMs. We want to **emphasize and illustrate** that our focus on linear models is both a *practical strength* and a *theoretical foundation*:
> > 1. **Practical importance**: Linear models remain the *workhorse of interpretable ML* (Rudin, 2019), causal inference, and scientific and practical applications across all fields.
> > 2. **Theoretical foundation**: Our tractable setting enables rigorous proofs that extend to broader classes. We *demonstrate this by sketching* how our EVT results (Theorems 1-2) extend to M-estimators.
> >
> > **Important caveat**: We view this extension as future work because *practical constraints* currently **limit actionability** – one of our core contributions. In this setting, the computational challenges (correctly emphasized by Reviewer PJQ8) are binding, which is *not the case in our current setting*. Hence, this would constrain the practical relevance and utility of the current paper.
> > ### Sketch: M-estimator Extension
> > M-estimators are defined by
> > $$
> > \hat{\theta} = \arg\min_{\theta \in \Theta} \sum_{i=1}^N \rho(z_i, \theta),
> > $$
> > where $z_i = (x_i, y_i)$ collects i.i.d. observations and $\rho$ is the loss function. Define:
> > - Score function: $\psi(z, \theta) = \nabla_\theta \rho(z, \theta),$
> > - Population Hessian: $H(\theta) = \mathbb{E}[\nabla^2_\theta \rho(z, \theta)],$
> > - Empirical Hessian: $\hat{H_N}(\theta) = N^{-1} \sum_{i=1}^N \nabla^2_\theta \rho(z_i, \theta),$
> >
> > and impose standard regularity conditions (twice differentiable loss, unique interior minimum, finite moments, positive definite Hessian) and tail conditions on the score function.
> >
> > **Key insight**: The influence on the M-estimator decomposes as:
> > $$
> >     \Delta(\mathcal{S}) \approx \hat{H_{-\mathcal{S}}}^{-1} \times p \sum_{s \in \mathcal{S}} \psi(z_s, \theta_{-\mathcal{S}}),
> > $$
> > where $p = \frac{1}{N - \lvert \mathcal{S} \rvert}$ and the remainder is $o_p(\lvert \mathcal{S} \rvert / N)$ and probabilistically negligible (tight for quadratic loss). This product structure is **conceptually identical** to our LS case. Thus, our EVT results (Theorem 1 and 2) extend naturally:
> > - **Theorem 1'** (M-estimator, Fréchet case): For $k$ fixed as $N \to \infty$ and polynomial-tailed scores, maximum influence converges to the Fréchet distribution.
> > - **Theorem 2'** (M-estimator, Gumbel cases): For either (a) $k$ fixed as $N \to \infty$ and exponential-tailed scores, or (b) growing set size $k_N = pN,$ maximum influence converges to the Gumbel distribution.
> >
> > The proofs follow the structure of our Lemmata: (i) asymptotic independence of numerator and denominator, (ii) EVD of the numerator, (iii) denominator convergence, and (iv) product inherits numerator's EVD.
> >
> > **Implications:** For example, maximum influence in **logistic regression** *always converges to the Gumbel case* due to the bounded score function.
> >
> > **Why This Remains Future Work:**
> > While the *theory* extends naturally, *practical implementations* for M-estimators face additional challenges. Unlike LS, where Proposition 1 gives closed-form
> > updates via Sherman-Morrison-Woodbury, M-estimators generally require iterative optimization (Newton-Raphson, IRLS) for each candidate set. Search over a limited number of candidate sets is tractable for LS (our applications handle $N > 10,000$ easily), but not in general. Developing practical tools for these extensions requires substantial future algorithmic development. We **strongly believe** that *establishing rigorous theoretical foundations* with *immediate practical applicability* – as our paper does for linear models – is an equally crucial contribution to this research program.
> > ### Summary
> > This extension demonstrates that our framework is **theoretically foundational** while highlighting the *practical importance* of our current contribution. Linear models provide the optimal setting for establishing rigorous statistical tests for influential sets: tractable enough for comprehensive theoretical and practical characterization, yet fundamental enough to guide extensions to broader model classes.
> >
> > Specifically, we believe this addresses:
> > - **Reviewer mjVQ's** concern about scope limitations to models "where the extreme
> >   value behavior of maximal influence can be characterized" – the theoretical pathway generalizes;
> > - **Reviewer 7xpC's** concern about limited theoretical contribution – we demonstrate the foundational nature with clear extensions;
> > - **Reviewer AdTk's** inquiry about application to M-estimation problems – we provide the theoretical extension while acknowledging practical constraints;
> > - **Reviewer PJQ8's** concern about limited scope and computational efficiency –
> >   computational tractability is a strength of our current setting, while extensions face the computational challenges correctly emphasized by the reviewer.

---

### Official Review · Reviewer_mbGm · 2025-10-30

**Soundness:** 3
**Presentation:** 3
**Contribution:** 2
**Rating:** 6
**Confidence:** 2

**Summary:**

This work introduces a rigorous hypothesis testing framework for influential sets on the coefficient in a univariate regression. It identifies the asymptotic distribution for the maximum influence of any such set for both fixed size and varying sized sets. These insights are then used to test whether the influence of a set is significant, meaning its influence is beyond that expected from natural variations in data. This framework is validated and applied in six case studies.

**Strengths:**

- The paper is generally well written, with clean and easy to follow figures.
- This work is original, presenting the first hypothesis testing framework for influential observations (to my knowledge).
- This work is well motivated. The problem of rigorous testing for influential observations is challenging, but has ramifications for a wide range of practical applications. This range is highlighted well by the variety of case studies in this paper. In particular, the question of how terrain ruggedness influences GDP is interesting. Broadly, the domains chosen for these case studies are well selected.
- The presented statistical theory is rigorous and validated through simulation studies.

**Weaknesses:**

- The presented framework only applies to univariate regression with positive coefficients. This restriction excludes many practical use cases, including simple extensions like multivariate regression. As such, the machine learning case studies presented in this work seem somewhat contrived, as machine learning practitioners tend to approach these datasets using multivariate regression.
- It seems like there is a risk of extreme multiple hypothesis testing problem here. For example, on the Boston housing dataset, six observations were selected as the most influential set. If these were selected after considering, say, the 1, 2, 3, 4, and 5 most suspicious observations, six hypothesis tests have already been performed (or at the very least, six effect sizes measured). In practice, it seems very tempting (and easy) to consider many potential influential sets, ultimately invalidating the conclusion or making the p-values large if a correction is applied.

**Questions:**

- In the equation before (1), should the first sum be over n \in [N] rather than not in S?
- It is stated that "we assume a univariate model with a positive coefficient". However, the machine learning case studies include cases with negative coefficients. Does this violate the theory, or am I missing something?

---

> ### Author Response · Authors · 2025-11-19
> **Summary Response**
>
> ## General Response
> We thank the reviewers for their constructive engagement with our work. We are encouraged by the consensus on theoretical rigor, novelty, and importance of addressing the long-standing challenge of assessing influential sets.
>
> **Core contribution.** Several reviewers asked us to clarify what our framework adds beyond existing influence quantification methods. The answer is fundamental: existing methods can identify influential sets but *cannot determine if their influence is excessive*.
> Prior work relies on ad-hoc heuristics (sign-flips, arbitrary thresholds like $2\sqrt{N}$, percentage rules) without theoretical foundation. Our contribution derives the distribution of maximum influence, enabling the first principled hypothesis tests to distinguish excessive influence from natural sampling variation. This transforms influential set analysis from heuristic sensitivity checking into rigorous statistical inference.
>
> Our revision addresses three main themes: (1) **scope and generalizability**, (2) **practical guidance and workflow**, and (3) **clarity of exposition**. We detail related revisions next, and follow with individual point-by-point responses to concerns raised by reviewers in the reply to this summary.
> ### Generalization beyond univariate OLS.
> We substantially expanded our theoretical framework.
> Proposition 1 now applies to *multivariate* least-squares estimators with arbitrary $P$-dimensional parameter vectors, encompassing both OLS ($\lambda = 0$) and ridge regression ($\lambda > 0$). Ridge regression ensures invertibility in high-dimensional settings where $P \gg N,$ covering practical ML applications.
> All *empirical applications* use the multivariate framework; the univariate case serves purely as *illustration* to build intuition. Full technical details appear in Appendix A1.
> ### Practical guidance and workflow.
> We enhanced practical guidance to address concerns about applicability:
>
> - **Convergence:** New simulations show reliable convergence at $N=100$  (and $N=20$ for light-tailed cases).
> - **Workflow:** We now cover the workflow before and after our test. *Before testing,* we clarify set selection and when multiple testing corrections apply. *After detection* of excessive influence, we provide guidance on investigation and responses.
> - **Why testing matters:** We clarified that our contribution provides theoretically-founded decision rules to replace ad-hoc heuristics. - Existing methods quantify influence; we enable principled decisions on whether that influence is excessive.
> ### Clarity and presentation.
> We improved exposition by adding background information on extreme value theory (including intuition and an illustrative figure), streamlining the influence function discussion and moved up key citations, fixed notational issues, and restructured for logical flow.

---

> > ### Author Response · Authors · 2025-11-19
> > **Individual Response to mbGm**
> >
> > ## Response to Reviewer mbGm
> > We thank the reviewer for the positive assessment of our originality, motivation, rigor, and presentation, and are encouraged by the recognition as "the first hypothesis testing framework for influential observations" with "ramifications for a wide range of practical applications." We address the two main concerns below.
> >
> > ### W1: Restriction to univariate regression with positive coefficients
> > We have substantially generalized our framework. The univariate, positive coefficient case was chosen for illustration to simplify exposition and build intuition.
> > The revised paper now features full generality: Proposition 1 applies to the multivariate case with arbitrary $P$-dimensional parameter vectors and optional regularization (ridge regression). The sign assumption is without loss of generality – it can be handled by adjusting the target function $\phi$ or flipping the sign of the corresponding feature in $X$.
> >
> > Empirical applications demonstrate the multivariate framework: the sparrow application includes multiple morphological features, while Communities & Crime includes over 100 socioeconomic and demographic variables.
> >
> > ### W2: Multiple hypothesis testing
> > This is an excellent and important point. We now address this in the subsections on extreme value theory and implementation. The multiple testing problem operates at two levels:
> >
> > 1. **Searching subsets:** For most influential sets of size $k,$ researchers consider $\binom{N}{k}$ possible subsets. Our distribution *already accounts for this* – this is precisely why we derive EVDs rather than appealing to a CLT. (The maximum behaves fundamentally differently than influence in general.) We test the *maximum* over all subsets, which is characterized by EVT, not individual subsets.
> > 2. **Searching scenarios:** If researchers test multiple distinct hypotheses (e.g., influence on different coefficients or different subset sizes), standard multiple testing corrections apply. We address a standard workflow where substantive concerns (e.g., the Seychelles) or heuristics (e.g., a sign-flip) identify one set of interest. For multiple scenarios, standard corrections should be applied.
> >
> > **Practical guidance:** We suggest distinguishing exploratory analysis (identifying sets) from confirmatory testing (our framework). Moreover, empirical applications often show $p \ll 0.01,$ which would remain significant even after multiple testing corrections.
> > There is an additional safety margin: If researchers use greedy algorithms (as we do; see Hu et al. for reasons why), there is a risk of not finding the *true* maximum. As discussed in response to Reviewer mjVQ, our test is *conservative* – we may fail to detect excessive influence (Type II error) but will not falsely claim it (Type I error control). This provides additional protection against some forms of multiple testing.
> > ### Questions
> > - **Typo in influence section:** That is correct – thank you for catching this! It is fixed in the revision.
> > - **Negative coefficients in applications:** Our initial presentation was unclear. Negative coefficients are accommodated by defining the target function appropriately or flipping the sign of the corresponding feature. Both ensure we test influence on the *magnitude* of the effect in the expected direction. The theory requires only that we fix a direction of interest – the actual sign is flexible. We have clarified this in the revision.
> >
> > ### Summary
> > The main concerns – univariate restriction and multiple testing – have been directly addressed through generalized theory, explicit discussion, and practical recommendations. Combined with the reviewer's recognition of our originality and rigor, we hope these substantial improvements lead to a more favorable assessment. The framework is now demonstrably applicable to the multivariate ML settings the reviewer correctly identifies as relevant use cases.

---

> > > ### Comment · Reviewer_mbGm · 2025-11-20
> > >
> > > I appreciate the authors' thorough response! The changes in the work have helped to address my primary concern (that the applicability of the theory was limited). I recognize the other reviewers' point that there is still room to broaden the scope, but believe the move to multivariate OLS with regularization is sufficient to make this a useful tool for practitioners.
> > >
> > > I find the acknowledgement of the potential need for a multiple testing correction satisfying.
> > >
> > > Because my primary weaknesses have been addressed, I am happy to raise my score to a 7, although I maintain my lower confidence score as this is not my primary area of research. My understanding is that this work offers a useful, rigorous tool for detecting undue influence in an area that has not had such a tool. As such, I believe it is ready to publish and share with the ICLR community.

---

> > > > ### Author Response · Authors · 2025-11-20
> > > >
> > > > Thank you for your engagement and consideration of our work! Your feedback has substantially strengthened the paper, and we appreciate your willingness to evaluate our work despite this not being your primary research area.
> > > >
> > > > We're glad that you agree we offer *a rigorous tool for detecting undue influence* – capturing precisely what we hoped to contribute to the ICLR community. Thank you again for your supportive assessment.

---

> > > > ### Author Response · Authors · 2025-11-26
> > > >
> > > > Dear reviewer mbGm,
> > > >
> > > > We wanted to follow up on your earlier response – the updated score doesn't appear to have been updated in the system yet. We would be very grateful if you could finalize this change when you have a moment, as it would significantly improve the chances of reaching the ICLR community.
> > > >
> > > > Thank you again for your support!

---

> > > > > ### Comment · Reviewer_mbGm · 2025-11-26
> > > > >
> > > > > I have updated my score accordingly -- it looks like only 2 point increments are available, so I have set the reported number as an 8 and made explicit that my intended rating is a 7. Thank you again for your detailed response, and continued efforts in addressing all reviewer concerns.

---

### Official Review · Reviewer_AdTk · 2025-10-31

**Soundness:** 2
**Presentation:** 3
**Contribution:** 2
**Rating:** 2
**Confidence:** 4

**Summary:**

This paper studies maximally influential data subsets in linear regression and proposes a statistical testing framework to determine when identified influential sets reflect excessive rather than natural sampling variability. The authors derive asymptotic extreme value distributions (Fréchet vs. Gumbel) for the maximum influence statistic under different scaling regimes of subset size, provide computation formulas for exact influence, and apply the approach to simulated and real datasets across multiple domains.

**Strengths:**

1. This paper tackles an important question on understanding the influential data subsets on the statistical estimator.
2. The presentation is easy to understand, with theoretical contributions.

**Weaknesses:**

1. The motivation for p-values in influential subset testing is not clearly justified. The main distinction from Broderick et al., (2021) is that this paper added a statistical significance test, but the paper does not convincingly demonstrate why this hypothesis-testing perspective materially improves decision-making in practice. For example, all the real data applications in this paper can be similarly done wth Broderick et al. (2020). I would suggest that the authors provide why the proposed approach yields better conclusions by using these p-values compared to existing influence quantification approaches.

2. The proposed approach is limited only to linear regression, while Broderick et al. (2021) can be applied to all M-estimation problems. Extensions to GLMs or modern models are only speculated about, yet the paper argues broadly about interpretability and fairness across ML. Can this approach be applied to all M-estimation problems?

Reference:
Broderick, T., Giordano, R., & Meager, R. (2020). An automatic finite-sample robustness metric: when can dropping a little data make a big difference?. arXiv preprint arXiv:2011.14999.

**Questions:**

1. See weaknesses 1 and 2
2. Would the hypothesis test disagree meaningfully with simpler leverage or leave-one-out metrics or Broderick et al. (2021)?
3. What are the real-world applications that this approach can benefit? I understand that people can use it to find the influential sets, but what should be the next steps?

---

> ### Author Response · Authors · 2025-11-19
> **Summary Response**
>
> ## General Response
> We thank the reviewers for their constructive engagement with our work. We are encouraged by the consensus on theoretical rigor, novelty, and importance of addressing the long-standing challenge of assessing influential sets.
>
> **Core contribution.** Several reviewers asked us to clarify what our framework adds beyond existing influence quantification methods. The answer is fundamental: existing methods can identify influential sets but *cannot determine if their influence is excessive*.
> Prior work relies on ad-hoc heuristics (sign-flips, arbitrary thresholds like $2\sqrt{N}$, percentage rules) without theoretical foundation. Our contribution derives the distribution of maximum influence, enabling the first principled hypothesis tests to distinguish excessive influence from natural sampling variation. This transforms influential set analysis from heuristic sensitivity checking into rigorous statistical inference.
>
> Our revision addresses three main themes: (1) **scope and generalizability**, (2) **practical guidance and workflow**, and (3) **clarity of exposition**. We detail related revisions next, and follow with individual point-by-point responses to concerns raised by reviewers in the reply to this summary.
> ### Generalization beyond univariate OLS.
> We substantially expanded our theoretical framework.
> Proposition 1 now applies to *multivariate* least-squares estimators with arbitrary $P$-dimensional parameter vectors, encompassing both OLS ($\lambda = 0$) and ridge regression ($\lambda > 0$). Ridge regression ensures invertibility in high-dimensional settings where $P \gg N,$ covering practical ML applications.
> All *empirical applications* use the multivariate framework; the univariate case serves purely as *illustration* to build intuition. Full technical details appear in Appendix A1.
> ### Practical guidance and workflow.
> We enhanced practical guidance to address concerns about applicability:
>
> - **Convergence:** New simulations show reliable convergence at $N=100$  (and $N=20$ for light-tailed cases).
> - **Workflow:** We now cover the workflow before and after our test. *Before testing,* we clarify set selection and when multiple testing corrections apply. *After detection* of excessive influence, we provide guidance on investigation and responses.
> - **Why testing matters:** We clarified that our contribution provides theoretically-founded decision rules to replace ad-hoc heuristics. - Existing methods quantify influence; we enable principled decisions on whether that influence is excessive.
> ### Clarity and presentation.
> We improved exposition by adding background information on extreme value theory (including intuition and an illustrative figure), streamlining the influence function discussion and moved up key citations, fixed notational issues, and restructured for logical flow.

---

> > ### Author Response · Authors · 2025-11-19
> > **Individual Response to Reviewer AdTk**
> >
> > ## Response to Reviewer AdTk
> > We thank the reviewer for engaging deeply with our work and raising important questions about its practical value. We agree the original manuscript communicated this poorly and believe our revision substantially addresses these concerns. We respectfully request reconsideration.
> >
> > ### W1: Why p-values?
> > This is a core question, and we have strengthened the paper to better address it. The fundamental issue is that *existing methods can identify influential sets*, but they *cannot determine if their influence is problematic*.
> >
> > Broderick et al. and related work quantify influence, but without knowledge of its distribution, they rely on *ad-hoc heuristics*:
> >
> > - **Sign-flips:** Any dataset with sufficient variation will have *some* subset that flips a coefficient's sign – even for ground-truth positive effects.
> > - **Loss of significance:** One can always find subsets crossing $p$-value thresholds.
> > - **Percentage thresholds:** Rules like "sets below 5% of data" are arbitrary and lack statistical foundation.
> >
> > There are two distinct questions: (1) How much influence does a set have? (quantified by existing methods); (2) Is that influence *excessive* or is it expected given natural sampling variation? (unaddressed). **Our contribution** is the first principled answer to the second question – by deriving the distribution of maximum influence we can meaningfully compare influence. This enables:
> >
> > - **Rigorous decisions:** test whether "influence is statistically incompatible with natural variation" (e.g., $p<0.001$) rather than relying on "causes a sign-flip" (always possible) or "is small enough" (arbitrary).
> > - **Preventing false alarms:** a 1% subset has substantial influence, but $p>0.05$ indicates that it stems from natural variation (Adult Income).
> > - **Catching true problems:** a 17-point set lacks a sign-flip, but has $p=0.019,$ indicating excessive influence (Law School).
> >
> > We have substantially revised the manuscript to articulate these distinctions throughout.
> >
> > ### W2: Scope – Linear regression vs. M-estimation
> >
> > We acknowledge that Broderick et al. achieve broad applicability via influence function approximations. However, this follows a fundamental trade-off:
> >
> > - **Their approach & limitation:** Broad applicability via first-order approximations for systematic underestimation of influence for sets and extreme cases (see Basu et al., 2020; Hu et al., 2024; Huang et al., 2025; Koh et al., 2019)
> > - **Our approach & limitation:** Exact influence computation and accurate asymptotic theory for least-squares (including multivariate and penalized)
> >
> > Why does this matter? Characterizing the distribution of maximum influence requires analytically tractable expressions and accurate influence computation. With approximations, errors compound precisely for the extreme influential sets we target and testing become misleading.
> >
> > ### Q: Would hypothesis tests disagree meaningfully with simpler metrics?
> >
> > Yes, fundamentally – our approach addresses a different question:
> >
> > 1. **Leverage scores:** Measure data geometry, not influence. A high-leverage point with zero residual, for instance, exerts no influence.
> > 2. **Leave-one-out (LOO):** Cannot capture *joint influence* – multiple observations can have small individual LOO influence but large collective influence through interaction. (See Kuschnig et al.'s Figure 1 for a clear example).
> > 3. **Broderick et al. heuristics:** Our test may disagree in both directions, as documented above.
> >
> > ### Q: Next steps after identifying excessive influence?
> > Excellent question. We have significantly expanded practical guidance, and recommend the following:
> >
> > 1. **Investigate mechanism:** Data quality (measurement errors), heterogeneity (subpopulations), unobserved confounding, or model misspecification (nonlinearity, missing interactions)
> > 2. **Handle appropriately:** Correct or exclude data errors with documentation; analyze heterogeneous subgroups separately; adjust model specification as needed
> > 3. **Report transparently:** Document the influential set and justify decisions with statistical evidence (e.g., "excluded due to excessive influence, p < 0.001") rather than arbitrary rules
> >
> > We explicitly warn against transformations that force alignment (winsorizing, trimming), as these may obscure genuine patterns. The key principle: excessive influence signals that something requires attention. Our framework detects this signal; domain expertise guides the response.
> >
> > ### Summary
> > Our contribution is *not* better influence quantification, but **principled statistical inference about influence**. We transform the question from "how much does this set influence the estimate?" (existing methods) to "is this influence statistically excessive?"
> >
> > The substantial revisions – expanded theory, detailed practical guidance, comparisons with existing methods, and clearer articulation – directly address your concerns, and we respectfully request reconsideration in light of these improvements.

---

> ### Author Response · Authors · 2025-11-26
> **Addressing scope concerns and M-estimators**
>
> ## Addressing Scope Concerns
> Multiple reviewers raised concerns about the scope of our framework, so we sketch the **extension of our results to M-estimators**, including logistic regression and GLMs. We want to **emphasize and illustrate** that our focus on linear models is both a *practical strength* and a *theoretical foundation*:
> 1. **Practical importance**: Linear models remain the *workhorse of interpretable ML* (Rudin, 2019), causal inference, and scientific and practical applications across all fields.
> 2. **Theoretical foundation**: Our tractable setting enables rigorous proofs that extend to broader classes. We *demonstrate this by sketching* how our EVT results (Theorems 1-2) extend to M-estimators.
>
> **Important caveat**: We view this extension as future work because *practical constraints* currently **limit actionability** – one of our core contributions. In this setting, the computational challenges (correctly emphasized by Reviewer PJQ8) are binding, which is *not the case in our current setting*. Hence, this would constrain the practical relevance and utility of the current paper.
> ### Sketch: M-estimator Extension
> M-estimators are defined by
> $$
> \hat{\theta} = \arg\min_{\theta \in \Theta} \sum_{i=1}^N \rho(z_i, \theta),
> $$
> where $z_i = (x_i, y_i)$ collects i.i.d. observations and $\rho$ is the loss function. Define:
> - Score function: $\psi(z, \theta) = \nabla_\theta \rho(z, \theta),$
> - Population Hessian: $H(\theta) = \mathbb{E}[\nabla^2_\theta \rho(z, \theta)],$
> - Empirical Hessian: $\hat{H_N}(\theta) = N^{-1} \sum_{i=1}^N \nabla^2_\theta \rho(z_i, \theta),$
>
> and impose standard regularity conditions (twice differentiable loss, unique interior minimum, finite moments, positive definite Hessian) and tail conditions on the score function.
>
> **Key insight**: The influence on the M-estimator decomposes as:
> $$
>     \Delta(\mathcal{S}) \approx \hat{H_{-\mathcal{S}}}^{-1} \times p \sum_{s \in \mathcal{S}} \psi(z_s, \theta_{-\mathcal{S}}),
> $$
> where $p = \frac{1}{N - \lvert \mathcal{S} \rvert}$ and the remainder is $o_p(\lvert \mathcal{S} \rvert / N)$ and probabilistically negligible (tight for quadratic loss). This product structure is **conceptually identical** to our LS case. Thus, our EVT results (Theorem 1 and 2) extend naturally:
> - **Theorem 1'** (M-estimator, Fréchet case): For $k$ fixed as $N \to \infty$ and polynomial-tailed scores, maximum influence converges to the Fréchet distribution.
> - **Theorem 2'** (M-estimator, Gumbel cases): For either (a) $k$ fixed as $N \to \infty$ and exponential-tailed scores, or (b) growing set size $k_N = pN,$ maximum influence converges to the Gumbel distribution.
>
> The proofs follow the structure of our Lemmata: (i) asymptotic independence of numerator and denominator, (ii) EVD of the numerator, (iii) denominator convergence, and (iv) product inherits numerator's EVD.
>
> **Implications:** For example, maximum influence in **logistic regression** *always converges to the Gumbel case* due to the bounded score function.
>
> **Why This Remains Future Work:**
> While the *theory* extends naturally, *practical implementations* for M-estimators face additional challenges. Unlike LS, where Proposition 1 gives closed-form
> updates via Sherman-Morrison-Woodbury, M-estimators generally require iterative optimization (Newton-Raphson, IRLS) for each candidate set. Search over a limited number of candidate sets is tractable for LS (our applications handle $N > 10,000$ easily), but not in general. Developing practical tools for these extensions requires substantial future algorithmic development. We **strongly believe** that *establishing rigorous theoretical foundations* with *immediate practical applicability* – as our paper does for linear models – is an equally crucial contribution to this research program.
> ### Summary
> This extension demonstrates that our framework is **theoretically foundational** while highlighting the *practical importance* of our current contribution. Linear models provide the optimal setting for establishing rigorous statistical tests for influential sets: tractable enough for comprehensive theoretical and practical characterization, yet fundamental enough to guide extensions to broader model classes.
>
> Specifically, we believe this addresses:
> - **Reviewer mjVQ's** concern about scope limitations to models "where the extreme
>   value behavior of maximal influence can be characterized" – the theoretical pathway generalizes;
> - **Reviewer 7xpC's** concern about limited theoretical contribution – we demonstrate the foundational nature with clear extensions;
> - **Reviewer AdTk's** inquiry about application to M-estimation problems – we provide the theoretical extension while acknowledging practical constraints;
> - **Reviewer PJQ8's** concern about limited scope and computational efficiency –
>   computational tractability is a strength of our current setting, while extensions face the computational challenges correctly emphasized by the reviewer.

---

> > ### Comment · Reviewer_AdTk · 2025-11-27
> >
> > I appreciate the authors’ detailed response. My concerns are resolved, and I now support the paper for ICLR. I would encourage the authors to include illustrative examples demonstrating when using a p-value in this task offers advantages over simpler approaches such as sign flipping or other influence-point removing methods.

---

> > > ### Author Response · Authors · 2025-12-01
> > >
> > > Many thanks for your thoughtful evaluation of our work and for engaging so openly with it!
> > > We're glad that our responses helped resolve your concerns, and we will prominently incorporate illustrative examples of the advantages of formal testing in the next revision.

---

### Official Review · Reviewer_7xpC · 2025-11-01

**Soundness:** 3
**Presentation:** 3
**Contribution:** 2
**Rating:** 4
**Confidence:** 3

**Summary:**

This paper examines the most influential sets problem for linear regression model from the hypothesis testing point of view, deriving the limiting distribution of the influence of most influential sets.

**Strengths:**

1. Novel theoretical results: I'm not aware of related literatures that address the limiting distribution of the maximum influence for linear regression model.
2. Conceptual clarity and motivation: The identified gap is indeed an important research question to address.

**Weaknesses:**

1. Practical guidance: Since the theory focuses on asymptotic behavior, it is unclear how many samples are needed to render the theory applicable.
2. Presentation: The clarity of the paper can be improved by carefully restructuring the sections. For instance, it seems like the presentation before Section 3.2 largely follows [1]. However, some parts are not necessarily used, for instance, the influence function. While I won't say this is to the extent of "plagiarism", however, I do think some careful adaptation to the current paper will be more appropriate.
3. Limited scope: This is my main concern. While I do think this paper studies an important research question, however, the scope and the result are limited. With the fact that the empirical experiments are also focusing on simple datasets and without significant empirical findings, I do not think I can suggest acceptance to the conference. This is only my opinion, and I'll leave the final judgment to the AC.


[1] Yuzheng Hu, Pingbang Hu, Han Zhao, and Jiaqi W. Ma. Most influential subset selection: Challenges, promises, and beyond.

**Questions:**

1. Typo: Line 118 states that $\epsilon=0$ recovers $\hat{\theta}$, which is think is not, as the left-hand side is summing over $n\notin \mathbb{S}$. Similarly, the claim for $\epsilon=-N^{-1}$ does not recover $\hat{\theta}_{-\mathbb{S}}$, but rather $\epsilon=0$.
2. Typo: Equation (1): It is *fine* to define influence as what it is, but it is far from correct to write $\hat{\theta}(\epsilon; \mathbb{S}) \approx \mathcal{I}(\mathbb{S})$. The correct interpretation is that, $\\hat{\\theta}\_{-\\emptyset} - \\hat{\\theta}\_{- \\mathbb{S}} \\approx \\epsilon \\mathcal{I} (\\mathbb{S})$.
3. Line 113 should reference the influence function paper by Koh & Liang, 2017.
4. To justify the contribution, I think it'll be beneficial to provide some critical (real-world/conceptual) examples that require hypothesis testing for the most influential sets problem.

---

> ### Author Response · Authors · 2025-11-19
> **Summary Response**
>
> ## General Response
> We thank the reviewers for their constructive engagement with our work. We are encouraged by the consensus on theoretical rigor, novelty, and importance of addressing the long-standing challenge of assessing influential sets.
>
> **Core contribution.** Several reviewers asked us to clarify what our framework adds beyond existing influence quantification methods. The answer is fundamental: existing methods can identify influential sets but *cannot determine if their influence is excessive*.
> Prior work relies on ad-hoc heuristics (sign-flips, arbitrary thresholds like $2\sqrt{N}$, percentage rules) without theoretical foundation. Our contribution derives the distribution of maximum influence, enabling the first principled hypothesis tests to distinguish excessive influence from natural sampling variation. This transforms influential set analysis from heuristic sensitivity checking into rigorous statistical inference.
>
> Our revision addresses three main themes: (1) **scope and generalizability**, (2) **practical guidance and workflow**, and (3) **clarity of exposition**. We detail related revisions next, and follow with individual point-by-point responses to concerns raised by reviewers in the reply to this summary.
> ### Generalization beyond univariate OLS.
> We substantially expanded our theoretical framework.
> Proposition 1 now applies to *multivariate* least-squares estimators with arbitrary $P$-dimensional parameter vectors, encompassing both OLS ($\lambda = 0$) and ridge regression ($\lambda > 0$). Ridge regression ensures invertibility in high-dimensional settings where $P \gg N,$ covering practical ML applications.
> All *empirical applications* use the multivariate framework; the univariate case serves purely as *illustration* to build intuition. Full technical details appear in Appendix A1.
> ### Practical guidance and workflow.
> We enhanced practical guidance to address concerns about applicability:
>
> - **Convergence:** New simulations show reliable convergence at $N=100$  (and $N=20$ for light-tailed cases).
> - **Workflow:** We now cover the workflow before and after our test. *Before testing,* we clarify set selection and when multiple testing corrections apply. *After detection* of excessive influence, we provide guidance on investigation and responses.
> - **Why testing matters:** We clarified that our contribution provides theoretically-founded decision rules to replace ad-hoc heuristics. - Existing methods quantify influence; we enable principled decisions on whether that influence is excessive.
> ### Clarity and presentation.
> We improved exposition by adding background information on extreme value theory (including intuition and an illustrative figure), streamlining the influence function discussion and moved up key citations, fixed notational issues, and restructured for logical flow.

---

> > ### Author Response · Authors · 2025-11-19
> > **Individual Response to Reviewer 7xpC**
> >
> > ## Response to Reviewer 7xpC
> > We thank the reviewer for recognizing the novelty of our theoretical results and the importance of the research question. We address each concern below and hope the substantial revisions lead to reconsideration of the rating.
> >
> > ### W3: Limited scope and empirical findings
> > We believe our revisions directly address this concern:
> >
> > **Theoretical scope:** Proposition 1 now provides a general multivariate framework that covers both OLS ($\lambda = 0$), ridge regression ($\lambda > 0$) and high-dimensional settings where $P \gg N.$
> > The univariate presentation serves purely as illustration; all empirical applications use the multivariate framework.
> >
> > **Empirical significance:** We respectfully suggest the empirical contributions are more substantial than initially apparent:
> >
> > - **Resolving contested findings:** "Blessing of Bad Geography" controversy illustrates our impact. Kuschnig et al. (2021) suspect but could not confirm excessive influence of the Seychelles. Our test definitively resolves this: $p < 0.001$ proves excessive influence, calling into question the original finding and transforming informed speculation into statistical fact.
> > - **Broad applicability:** Every study using influential set methods (Broderick et al., 2021; Kuschnig et al., 2021; Freund & Hopkins, 2023, etc.) faces the same problem: they quantify influence but cannot determine if that influence is excessive. Our framework provides the missing statistical foundation for this widely-used class of methods.
> > - **Evaluating heuristics:** We can assess longstanding rules like the $2\sqrt{N},$ showing when and how they fail – with immediate practical implications for how practitioners assess influence.
> >
> > We have strengthened the manuscript to articulate these contributions more clearly throughout.
> >
> > ### W1: Sample size requirements
> > Our improved simulations directly address this.
> > Theory applies reliably at $N = 100$ for varied distributions, and at $N = 20$ for light-tailed cases. Heavy-tailed and mixed cases converge marginally slower, but hold for $N \geq 100.$
> > These sample sizes cover the vast majority of practical datasets; we acknowledge further investigation of convergence as important future work in the Limitations section.
> >
> > ### W2: Presentation and background
> > We have restructured the Background section accordingly. While influence functions are the dominant framework in the literature and warrant brief coverage for context, we have: (1) streamlined the discussion, (2) moved key citations earlier (Koh & Liang, 2017; Hu et al., 2024), (3) explicitly contrasted approaches and explained why exactness is necessary (approximations systematically underestimate set influence), and (4) fixed notational issues per your suggestions.
> > The material now clearly serves as *motivation*, explaining where existing approaches fail.
> >
> > ### Questions
> > - **Line 118 & Equation (1):** Corrected, thank you.
> > - **Koh & Liang citation:** Moved to first mention of influence functions.
> > - **Real-world examples:** The paper provides several: replication studies (when sets overturn findings), data cleaning (exclusion decisions), fairness auditing (distinguishing bias from sampling variation), clinical trials (robustness of treatment effects).
> >
> > ### Summary
> > These improvements – generalized theory, convergence details, and clarified contributions – directly address concerns about scope and impact. Our framework provides the first principled solution to a fundamental problem in interpretable ML: determining when influence is excessive rather than natural.

---

> ### Comment · Reviewer_7xpC · 2025-11-19
>
> Thanks for the responses. I have one follow-up for W1:
>
> From my understanding, in the setting you were considering (i.e., OLS with the target function being one of the parameters), a closed-form for the exact set influence can also be derived in Proposition 1, similar to Hu et al., 2024 (Proposition 3.2). This is largely due to the simplicity of the problem settings. Do you think in this case, it is possible to get finite-sample guarantees? Or, for instance, can you derive the rate of convergence?
>
> I don't necessarily consider this a weakness or a must-have, just that it might be possible to extend further and strengthen the submission.
>
> For the other weaknesses, the theoretical contribution after the generalization is still limited. However, since I have not yet checked all the details for the new draft (which has gone through a somewhat major revision), I have lowered my confidence.

---

> > ### Author Response · Authors · 2025-11-19
> >
> > Thank you for the thoughtful follow-up question and for engaging with our revisions!
> >
> > **On finite-sample guarantees:** You raise an excellent point. Since we can derive exact set influence, obtaining finite-sample convergence rates for the EVD is feasible in principle. The rate will depend on the residuals and features so additional assumptions will be necessary. Adapting existing EVT results to our product-of-sums structure requires non-trivial extensions. We agree this strengthens the contribution and note it as future work.
> >
> > **On theoretical scope:** While our is focused on linear models, this setting is ubiquitous in ML interpretability and our framework provides the first rigorous solution to a 45-year-old problem (Cook, 1979), with clear extensions to other settings.
> > The gap that we address has persisted since Cook (1979), and our contribution can help push the vibrant literature on influential sets (Broderick et al., Hu et al., and many more) into empirical practice.
> >
> > We appreciate your willingness to reconsider given the major revisions and welcome any additional feedback.

---

> > ### Author Response · Authors · 2025-11-24
> > **Summary of Changes**
> >
> > Dear reviewer,
> >
> > As you correctly point out – the manuscript has undergone significant revision to improve clarity, structure, and scope while preserving all core results.
> > Since the *submission system did not generate a track-changes document*, we provide a **concise summary of changes below** (leveraging the source files available to us) to help facilitate your review:
> >
> > ## Changes
> > **Summary of Changes**: Improved structure, clarity, and scope – no changes to core theory. The revised manuscript strengthens presentation and expands practical guidance.
> > ### Key Changes
> > 1. **Streamlined Structure and Motivation**
> >     - Background section (Sec 2) provides **clearer progression**: formal setup -> influence function (sharpened) -> exact influence -> introduction of extreme value theory (new) -> linear setting (moved from Sec 3; clarified notation on population vs. sample quantities)
> >     - **New Figure 1** illustrates why EVT is necessary (maximization over subsets creates extreme value behavior) and which distributions emerge (Gumbel vs. Fréchet)
> > 2. **Sharper Framing**
> >     - Consistently frames the research question as distinguishing **"excessive" influence** from *natural sampling variation* (e.g., in the Introduction, Seychelles geography application, and for recommendations)
> > 3. **Generalized Theoretical Scope**
> >     - *Proposition 1* now explicitly covers **multivariate ridge regression** with optional penalization parameter $\lambda = 0$, with earlier univariate OLS as the special case
> >     - Previous version only alluded to this generalization – it is now stated clearly and elaborated upon in the Appendix (Reviewer PJQ8 provides another more elegant proof)
> > 4. **Enhanced Practical Guidance**
> >     - **Implementation** (Sec 3.3) adds discussion of set selection procedures and streamlined headings
> >     - **Discussion** (Sec 5) now includes explicit **practical recommendations**: guidance on investigating mechanisms behind influential sets, appropriate handling strategies, and cautions against data transformations that may obscure
> > 5. **Expanded Empirical Validation**
> >     - **Simulation study** investigates finite-sample convergence, demonstrating reliable performance at $N \geq 50$
> >     - **New Figure 3** visualizes convergence scenarios
> > 6. **Minor improvements**
> >     - Streamlined section headers, fixed typos, improved notation consistency throughout, improved citation placement

---

> > ### Author Response · Authors · 2025-11-26
> > **Addressing Scope Concerns**
> >
> > ## Addressing Scope Concerns
> > Multiple reviewers raised concerns about the scope of our framework, so we sketch the **extension of our results to M-estimators**, including logistic regression and GLMs. We want to **emphasize and illustrate** that our focus on linear models is both a *practical strength* and a *theoretical foundation*:
> > 1. **Practical importance**: Linear models remain the *workhorse of interpretable ML* (Rudin, 2019), causal inference, and scientific and practical applications across all fields.
> > 2. **Theoretical foundation**: Our tractable setting enables rigorous proofs that extend to broader classes. We *demonstrate this by sketching* how our EVT results (Theorems 1-2) extend to M-estimators.
> >
> > **Important caveat**: We view this extension as future work because *practical constraints* currently **limit actionability** – one of our core contributions. In this setting, the computational challenges (correctly emphasized by Reviewer PJQ8) are binding, which is *not the case in our current setting*. Hence, this would constrain the practical relevance and utility of the current paper.
> > ### Sketch: M-estimator Extension
> > M-estimators are defined by
> > $$
> > \hat{\theta} = \arg\min_{\theta \in \Theta} \sum_{i=1}^N \rho(z_i, \theta),
> > $$
> > where $z_i = (x_i, y_i)$ collects i.i.d. observations and $\rho$ is the loss function. Define:
> > - Score function: $\psi(z, \theta) = \nabla_\theta \rho(z, \theta),$
> > - Population Hessian: $H(\theta) = \mathbb{E}[\nabla^2_\theta \rho(z, \theta)],$
> > - Empirical Hessian: $\hat{H_N}(\theta) = N^{-1} \sum_{i=1}^N \nabla^2_\theta \rho(z_i, \theta),$
> >
> > and impose standard regularity conditions (twice differentiable loss, unique interior minimum, finite moments, positive definite Hessian) and tail conditions on the score function.
> >
> > **Key insight**: The influence on the M-estimator decomposes as:
> > $$
> >     \Delta(\mathcal{S}) \approx \hat{H_{-\mathcal{S}}}^{-1} \times p \sum_{s \in \mathcal{S}} \psi(z_s, \theta_{-\mathcal{S}}),
> > $$
> > where $p = \frac{1}{N - \lvert \mathcal{S} \rvert}$ and the remainder is $o_p(\lvert \mathcal{S} \rvert / N)$ and probabilistically negligible (tight for quadratic loss). This product structure is **conceptually identical** to our LS case. Thus, our EVT results (Theorem 1 and 2) extend naturally:
> > - **Theorem 1'** (M-estimator, Fréchet case): For $k$ fixed as $N \to \infty$ and polynomial-tailed scores, maximum influence converges to the Fréchet distribution.
> > - **Theorem 2'** (M-estimator, Gumbel cases): For either (a) $k$ fixed as $N \to \infty$ and exponential-tailed scores, or (b) growing set size $k_N = pN,$ maximum influence converges to the Gumbel distribution.
> >
> > The proofs follow the structure of our Lemmata: (i) asymptotic independence of numerator and denominator, (ii) EVD of the numerator, (iii) denominator convergence, and (iv) product inherits numerator's EVD.
> >
> > **Implications:** For example, maximum influence in **logistic regression** *always converges to the Gumbel case* due to the bounded score function.
> >
> > **Why This Remains Future Work:**
> > While the *theory* extends naturally, *practical implementations* for M-estimators face additional challenges. Unlike LS, where Proposition 1 gives closed-form
> > updates via Sherman-Morrison-Woodbury, M-estimators generally require iterative optimization (Newton-Raphson, IRLS) for each candidate set. Search over a limited number of candidate sets is tractable for LS (our applications handle $N > 10,000$ easily), but not in general. Developing practical tools for these extensions requires substantial future algorithmic development. We **strongly believe** that *establishing rigorous theoretical foundations* with *immediate practical applicability* – as our paper does for linear models – is an equally crucial contribution to this research program.
> > ### Summary
> > This extension demonstrates that our framework is **theoretically foundational** while highlighting the *practical importance* of our current contribution. Linear models provide the optimal setting for establishing rigorous statistical tests for influential sets: tractable enough for comprehensive theoretical and practical characterization, yet fundamental enough to guide extensions to broader model classes.
> >
> > Specifically, we believe this addresses:
> > - **Reviewer mjVQ's** concern about scope limitations to models "where the extreme
> >   value behavior of maximal influence can be characterized" – the theoretical pathway generalizes;
> > - **Reviewer 7xpC's** concern about limited theoretical contribution – we demonstrate the foundational nature with clear extensions;
> > - **Reviewer AdTk's** inquiry about application to M-estimation problems – we provide the theoretical extension while acknowledging practical constraints;
> > - **Reviewer PJQ8's** concern about limited scope and computational efficiency –
> >   computational tractability is a strength of our current setting, while extensions face the computational challenges correctly emphasized by the reviewer.

---

### Official Review · Reviewer_mjVQ · 2025-11-01

**Soundness:** 3
**Presentation:** 3
**Contribution:** 3
**Rating:** 6
**Confidence:** 3

**Summary:**

This paper investigates the problem of assess the statistical significance of the most influential sets in OLS and proposes the first testing framework by deriving extreme-value limits for the maximum subset influence and turning them into calibrated p-values.

The proposed method is applied to several existing datasets in economics, biology, and machine learning benchmarks, demonstrating that some datasets indeed contain outliers with excessive influence while the influence of most influential subsets in some other datasets are not significantly excessive than natural sampling variation.

**Strengths:**

- The problem of testing most influential subsets is novel and well-motivated.
- The proposed framework is theoretically sound.
- The proposed approach may find broad applications in scientific domains that relies on OLS for data analysis.

**Weaknesses:**

- The proposed framework is limited to OLS.
- The block-maxima MLE used in the proposed approach suffers from a bias.
- The theory is only applicable to the low dim regime.

Minor: this work may be a better fit to statistics or economic venues than machine learning ones.

**Questions:**

How does a failure to identify the most influential subset impact the estimated p-value?

---

> ### Author Response · Authors · 2025-11-19
> **Summary Response**
>
> ## General Response
> We thank the reviewers for their constructive engagement with our work. We are encouraged by the consensus on theoretical rigor, novelty, and importance of addressing the long-standing challenge of assessing influential sets.
>
> **Core contribution.** Several reviewers asked us to clarify what our framework adds beyond existing influence quantification methods. The answer is fundamental: existing methods can identify influential sets but *cannot determine if their influence is excessive*.
> Prior work relies on ad-hoc heuristics (sign-flips, arbitrary thresholds like $2\sqrt{N}$, percentage rules) without theoretical foundation. Our contribution derives the distribution of maximum influence, enabling the first principled hypothesis tests to distinguish excessive influence from natural sampling variation. This transforms influential set analysis from heuristic sensitivity checking into rigorous statistical inference.
>
> Our revision addresses three main themes: (1) **scope and generalizability**, (2) **practical guidance and workflow**, and (3) **clarity of exposition**. We detail related revisions next, and follow with individual point-by-point responses to concerns raised by reviewers in the reply to this summary.
> ### Generalization beyond univariate OLS.
> We substantially expanded our theoretical framework.
> Proposition 1 now applies to *multivariate* least-squares estimators with arbitrary $P$-dimensional parameter vectors, encompassing both OLS ($\lambda = 0$) and ridge regression ($\lambda > 0$). Ridge regression ensures invertibility in high-dimensional settings where $P \gg N,$ covering practical ML applications.
> All *empirical applications* use the multivariate framework; the univariate case serves purely as *illustration* to build intuition. Full technical details appear in Appendix A1.
> ### Practical guidance and workflow.
> We enhanced practical guidance to address concerns about applicability:
>
> - **Convergence:** New simulations show reliable convergence at $N=100$  (and $N=20$ for light-tailed cases).
> - **Workflow:** We now cover the workflow before and after our test. *Before testing,* we clarify set selection and when multiple testing corrections apply. *After detection* of excessive influence, we provide guidance on investigation and responses.
> - **Why testing matters:** We clarified that our contribution provides theoretically-founded decision rules to replace ad-hoc heuristics. - Existing methods quantify influence; we enable principled decisions on whether that influence is excessive.
> ### Clarity and presentation.
> We improved exposition by adding background information on extreme value theory (including intuition and an illustrative figure), streamlining the influence function discussion and moved up key citations, fixed notational issues, and restructured for logical flow.

---

> > ### Author Response · Authors · 2025-11-19
> > **Individual Response to Reviewer mjVQ**
> >
> > ## Response to Reviewer mjVQ
> > We thank the reviewer for the positive assessment of our work's novelty, theoretical soundness, and broad applications. We address each concern below.
> > ### W1 & W3: Limitations (OLS, low-dimensional)
> > We have substantially generalized our framework beyond the 1-D illustrative case. The new Proposition 1 (with full proof in Appendix A1) applies to **multivariate** least-squares estimators with arbitrary $P$-dimensional coefficients, including **ridge regression** ($\lambda > 0$).
> > The univariate case now serves purely as pedagogical illustration.
> >
> > Notably, our results explicitly accommodate **high-dimensional** settings where $P \gg N:$ ridge regression ensures invertibility while preserving our theoretical results. This covers many practical ML settings where regularization is standard practice.
> >
> > ### W2: Bias in block-maxima MLE
> > We agree this is an important practical consideration. We address it in three ways:
> > 1. **Bias correction:** For the crucial location parameter, we apply a bias correction $\tilde{a} = \hat{a} + b\log(M).$
> > 2. **Empirical validation:** Simulation results quantify the remaining bias, demonstrating that scale bias has minimal impact on $p$-values and testing decisions.
> > 3. **Future work:** We acknowledge improved estimators as important future work that would enhance practical performance.
> >
> > ### Q: Failing to identify the true most influential set
> > This is an excellent question – our test remains valid (controls Type I error) but loses power.
> >
> > If the given set $\tilde{\mathcal{S}}$ with influence $\tilde{\delta}$ is not truly the most influential set, then $\tilde{\delta} \leq \Delta^{\max}$ by definition.
> > Our test is then based on $p = \Pr(\Delta^{\max} \geq \tilde{\delta}),$ and – depending on the inequality's magnitude – will be less likely to reject the null hypothesis, losing power but maintaining validity.
> > As a result, the test is *conservative* in this setting: it may fail to reject (false negative) but will not incorrectly flag excessive influence (false positives are controlled). Inference for the given set is valid, but the true most influential set remains hidden and its excessive influence undetected.
> >
> > ### Minor: Venue appropriateness
> > We respectfully disagree. While our theoretical tools draw from statistics and applications span domains, the *problem* is fundamentally about interpretable and robust machine learning. Influential sets are central to fairness, interpretability, and robustness. Minuscule sets are known to impact even large language models (Souly et al., 2025).

---

> > > ### Comment · Reviewer_mjVQ · 2025-11-24
> > >
> > > I appreciate the reviewer's response. The generalization of the framework to ridge regression and high dim looks nice. I also appreciate the clarification regarding my question. I think this is overall a solid work. But this testing framework is only applicable to models where the extreme value behavior of maximal influence can be characterized, which limits its scope. I'm already positive in my original review and will maintain my rating.

---

> > > > ### Author Response · Authors · 2025-11-26
> > > >
> > > > Many thanks for your response and positive rating!
> > > >
> > > > We try and address your remaining concern regarding scope in the other comment.

---

> ### Author Response · Authors · 2025-11-25
> **Addressing Scope Concerns**
>
> ## Addressing Scope Concerns
> Multiple reviewers raised concerns about the scope of our framework, so we sketch the **extension of our results to M-estimators**, including logistic regression and GLMs. We want to **emphasize and illustrate** that our focus on linear models is both a *practical strength* and a *theoretical foundation*:
> 1. **Practical importance**: Linear models remain the *workhorse of interpretable ML* (Rudin, 2019), causal inference, and scientific and practical applications across all fields.
> 2. **Theoretical foundation**: Our tractable setting enables rigorous proofs that extend to broader classes. We *demonstrate this by sketching* how our EVT results (Theorems 1-2) extend to M-estimators.
>
> **Important caveat**: We view this extension as future work because *practical constraints* currently **limit actionability** – one of our core contributions. In this setting, the computational challenges (correctly emphasized by Reviewer PJQ8) are binding, which is *not the case in our current setting*. Hence, this would constrain the practical relevance and utility of the current paper.
> ### Sketch: M-estimator Extension
> M-estimators are defined by
> $$
> \hat{\theta} = \arg\min_{\theta \in \Theta} \sum_{i=1}^N \rho(z_i, \theta),
> $$
> where $z_i = (x_i, y_i)$ collects i.i.d. observations and $\rho$ is the loss function. Define:
> - Score function: $\psi(z, \theta) = \nabla_\theta \rho(z, \theta),$
> - Population Hessian: $H(\theta) = \mathbb{E}[\nabla^2_\theta \rho(z, \theta)],$
> - Empirical Hessian: $\hat{H_N}(\theta) = N^{-1} \sum_{i=1}^N \nabla^2_\theta \rho(z_i, \theta),$
>
> and impose standard regularity conditions (twice differentiable loss, unique interior minimum, finite moments, positive definite Hessian) and tail conditions on the score function.
>
> **Key insight**: The influence on the M-estimator decomposes as:
> $$
>     \Delta(\mathcal{S}) \approx \hat{H_{-\mathcal{S}}}^{-1} \times p \sum_{s \in \mathcal{S}} \psi(z_s, \theta_{-\mathcal{S}}),
> $$
> where $p = \frac{1}{N - \lvert \mathcal{S} \rvert}$ and the remainder is $o_p(\lvert \mathcal{S} \rvert / N)$ and probabilistically negligible (tight for quadratic loss). This product structure is **conceptually identical** to our LS case. Thus, our EVT results (Theorem 1 and 2) extend naturally:
> - **Theorem 1'** (M-estimator, Fréchet case): For $k$ fixed as $N \to \infty$ and polynomial-tailed scores, maximum influence converges to the Fréchet distribution.
> - **Theorem 2'** (M-estimator, Gumbel cases): For either (a) $k$ fixed as $N \to \infty$ and exponential-tailed scores, or (b) growing set size $k_N = pN,$ maximum influence converges to the Gumbel distribution.
>
> The proofs follow the structure of our Lemmata: (i) asymptotic independence of numerator and denominator, (ii) EVD of the numerator, (iii) denominator convergence, and (iv) product inherits numerator's EVD.
>
> **Implications:** For example, maximum influence in **logistic regression** *always converges to the Gumbel case* due to the bounded score function.
>
> **Why This Remains Future Work:**
> While the *theory* extends naturally, *practical implementations* for M-estimators face additional challenges. Unlike LS, where Proposition 1 gives closed-form
> updates via Sherman-Morrison-Woodbury, M-estimators generally require iterative optimization (Newton-Raphson, IRLS) for each candidate set. Search over a limited number of candidate sets is tractable for LS (our applications handle $N > 10,000$ easily), but not in general. Developing practical tools for these extensions requires substantial future algorithmic development. We **strongly believe** that *establishing rigorous theoretical foundations* with *immediate practical applicability* – as our paper does for linear models – is an equally crucial contribution to this research program.
> ### Summary
> This extension demonstrates that our framework is **theoretically foundational** while highlighting the *practical importance* of our current contribution. Linear models provide the optimal setting for establishing rigorous statistical tests for influential sets: tractable enough for comprehensive theoretical and practical characterization, yet fundamental enough to guide extensions to broader model classes.
>
> Specifically, we believe this addresses:
> - **Reviewer mjVQ's** concern about scope limitations to models "where the extreme
>   value behavior of maximal influence can be characterized" – the theoretical pathway generalizes;
> - **Reviewer 7xpC's** concern about limited theoretical contribution – we demonstrate the foundational nature with clear extensions;
> - **Reviewer AdTk's** inquiry about application to M-estimation problems – we provide the theoretical extension while acknowledging practical constraints;
> - **Reviewer PJQ8's** concern about limited scope and computational efficiency –
>   computational tractability is a strength of our current setting, while extensions face the computational challenges correctly emphasized by the reviewer.

---

### Author Response · Authors · 2025-12-01
**Discussion Summary for the Area Chair**

# AC Summary
We sincerely thank the reviewers for their constructive feedback and active engagement. This comment summarizes their concerns, our rebuttals, and how the discussion evolved.

**Paper summary:** We derive the extreme value distribution of maximum influence for least-squares estimators, providing the first principled framework to test whether the impact of a *most influential set* is excessive or compatible with natural sampling variation.

**Discussion outcome:** After rebuttal and discussion, **most reviewers express clear support for publication**, and *none are strongly opposed*. Initial concerns about scope, practical applicability, and clarity have been substantially addressed.
## Reviewer Discussion
### mjVQ
- Initially supported publication, with concerns about limitations to low-dimensional OLS.
- We generalized the framework beyond the univariate illustration to *multivariate and penalized LS*.
- The reviewer acknowledges our rebuttal and **maintains their positive rating**:
> [...] The generalization of the framework to ridge regression and high dim looks nice. [...] I think this is overall a solid work [...]
### 7xpC
- Highlighted the novelty and the “important research question to address,” but expressed concerns about: (i) *limited scope*, (ii) practical relevance in *small samples*, and (iii) *presentation* (typos, structure, reference placement).
- We (iii) corrected presentation issues and *reworked the background section*, (ii) *added simulations and a figure* showing EVD predictions already align well at $N = 100,$ and (i) made the full *multivariate and penalized LS framework explicit* and sketched *extensions to M-estimators*.
- The reviewer noted they **needed time to check the revised draft** (“since I have not yet checked all the details for the new draft”); their *final response was pending* when the discussion was frozen.
### AdTk
- Found the research question as “important,” but had *initial concerns* about: (i) the **motivation for testing** whether influential sets are excessive, (ii) differences to common *heuristics*, and (iii) extensibility to *M-estimators*.
- We (i) clarified our contribution is to **distinguish excessive from expected influence**, turning heuristic rules into formal tests, (ii) discussed concrete situations where *our principled test and heuristics disagree*, and (iii) sketched extensions to M-estimators / GLMs.
- The reviewer now explicitly **supports acceptance**:
> I appreciate the authors’ detailed response. My concerns are resolved, and I now support the paper for ICLR. [...]
### mbGm
- Supported acceptance from the start, stressing originality and practical relevance, but had concerns about (i) the *limitation to univariate OLS* and (ii) *multiple testing* over influential sets.
- We (i) generalized the theory to full *multivariate and penalized LS framework*, and (ii) explained how our extreme value analysis addresses the combinatorial dimension of testing *the most influential set*, and discuss corrections when assessing multiple such sets.
- The reviewer **further raised their score**, stating that:
> Because my primary weaknesses have been addressed, I am happy to raise my score [...]. My understanding is that this work offers a useful, rigorous tool for detecting undue influence in an area that has not had such a tool. As such, I believe it is ready to publish and share with the ICLR community.
### PJQ8
- Praised the theoretical framework and the empirical evaluation across domains, but raised concerns about: (i) *restriction to 1D OLS*, (ii) the finite-population perspective of *fixed data*, and  (iii) *exposition of EVT*. They were “willing to reconsider my rating if the authors can provide a clear and technically plausible pathway for extending their theoretical framework beyond the 1D OLS setting.”
- We (iii) added an explicit introduction and illustration of EVT in the reworked background section, (ii) elaborated on the frequentist perspective, and (i) extended the framework to full *multivariate and penalized LS*.
- The reviewer provided an elegant alternative proof for Proposition 1, and, after discussion, **raised their score**, while maintaining reservations about computational efficiency and scope.
- We clarified that computational efficiency is not limiting in our setting and highlighted the *foundational nature* of our theoretical contribution, sketching extension to M-estimators / GLMs. The reviewer *did not respond further* before the discussion was frozen.
## Summary
Across the discussion, reviewers broadly agree that the paper: (i) tackles a *long-standing and practically important problem*, (ii) provides *rigorous theory for maximal influence*, and (iii) offers *concrete tools and guidance* for assessing excessive influence.
Given the final reviewer positions – **explicit support from three reviewers** and a non-negative stance from the rest – we believe the paper now meets the bar for acceptance at ICLR.

---

### Meta-Review · Area_Chair_x351 · 2025-12-17

**Summary:**

This paper develops a principled hypothesis testing framework for most influential sets in least-squares regression, deriving extreme value distributions (Frechet for constant-size sets with heavy-tailed data; Gumbel for growing sets or light tails) that enable rigorous tests to distinguish excessive influence from natural sampling variation.

The main contribution of the paper is conceptually novel and interesting, providing the first statistically principled method for identifying excessively influential data subsets rather than relying on heuristic thresholds. The primary strengths include addressing a practically important problem that has lacked principled solutions for long—existing influence quantification methods rely on ad-hoc heuristics without theoretical foundation for determining when influence is excessive versus expected from natural variation. The work provides rigorous theory deriving exact influence formulas and identifying extreme value distributions of maximal influence, enabling the first principled hypothesis tests that transform influential set analysis from heuristic sensitivity checking into rigorous statistical inference.

One common concern among multiple reviewers is the limited scope restricted to 1D OLS regression. During the rebuttal phase, the authors substantially improved presentation, clarified positioning relative to prior work, and extended the original 1D OLS setting to multivariate regression with ridge regularization. This is significant. Given the potential impact of the developed framework, I recommend acceptance, while encouraging the authors to follow the reviewers' suggestion to simplify the proof of Proposition 1, and to accurately contextualize and frame its contributions and limitations in the final version.

**Reviewer Concerns:**

The most important concerns about the scope and setting of the proposed framework have been addressed during the rebuttal. The remaining ones are minor

**Reviewer Scores:**

two of the reviewers remain the rating at 4, whereas the other three support acceptance. I appreciate the core contribution of this work, and I believe it can inspire many follow-up studies.

---

### Decision · Program_Chairs · 2026-01-26

Accept (Poster)